# PaLI: A Jointly-Scaled Multilingual Language-Image Model

**Xi Chen**,* **Xiao Wang,  Soravit Changpinyo,  AJ Piergiovanni,  Piotr Padlewski**
**Daniel Salz,  Sebastian Goodman,  Adam Grycner,  Basil Mustafa,  Lucas Beyer**
**Alexander Kolesnikov,  Joan Puigcerver,  Nan Ding,  Keran Rong,  Hassan Akbari**
**Gaurav Mishra,  Linting Xue,  Ashish Thapliyal,  James Bradbury,  Weicheng Kuo**
**Mojtaba Seyedhosseini,  Chao Jia,  Burcu Karagol Ayan,  Carlos Riquelme**
**Andreas Steiner,  Anelia Angelova,  Xiaohua Zhai,  Neil Houlsby,  Radu Soricut**
Google Research

## ABSTRACT

Effective scaling and a flexible task interface enable large language models to excel at many tasks. We present **PaLI** (**Pa**thways **L**anguage and **I**mage model), a model that extends this approach to the joint modeling of language and vision. PaLI generates text based on visual and textual inputs, and with this interface performs many vision, language, and multimodal tasks, in many languages. To train PaLI, we make use of large pre-trained encoder-decoder language models and Vision Transformers (ViTs). This allows us to capitalize on their existing capabilities and leverage the substantial cost of training them. We find that joint scaling of the vision and language components is important. Since existing Transformers for language are much larger than their vision counterparts, we train a large, 4-billion parameter ViT (ViT-e) to quantify the benefits from even larger-capacity vision models. To train PaLI, we create a large multilingual mix of pre-training tasks, based on a new image-text training set containing 10B images and texts in over 100 languages. PaLI achieves state-of-the-art in multiple vision and language tasks (such as captioning, visual question-answering, scene-text understanding), while retaining a simple, modular, and scalable design.

## 1   INTRODUCTION

Increasing neural network capacity has been a successful trend in the modeling of language and vision tasks. On the language side, models such as T5 (Raffel et al., 2020), GPT-3 (Brown et al., 2020), Megatron-Turing (Shoeybi et al., 2019), GLaM (Du et al., 2022), Chinchilla (Hoffmann et al., 2022), and PaLM (Chowdhery et al., 2022) have shown significant advantages from training large Transformers on large amounts text data. On the vision side, CNNs (Mahajan et al., 2018; Huang et al., 2019; Kolesnikov et al., 2020), Vision Transformers (Dosovitskiy et al., 2021), and other models (Tolstikhin et al., 2021; Riquelme et al., 2021) have seen similar benefits from scale (Zhai et al., 2022a), albeit to a lesser extent than in language. Language-and-vision modeling has followed a similar trend, e.g., SimVLM (Wang et al., 2021), Florence (Yuan et al., 2021), CoCa (Yu et al., 2022), GIT (Wang et al., 2022a), BEiT-3 (Wang et al., 2022c), and Flamingo (Alayrac et al., 2022).

We introduce PaLI, a model that performs image-only, language-only, and image+language tasks across many languages, using a single "image-and-text to text" interface. A key characteristic of PaLI is a more balanced parameter share between the language and vision components, with more capacity to the vision backbone yielding large gains in performance. Another key ingredient to PaLI is the reuse of large unimodal backbones for language and vision modeling, in order to transfer existing capabilities and reduce training cost. On the language side, we reuse the 13B-parameter model mT5-XXL (Xue et al., 2021), which already packages language understanding and generation capabilities. We show that these capabilities are maintained and extended into a multimodal setting. On the vision side, in addition to reusing the 2B-parameter ViT-G model (Zhai et al., 2022a), we

---

*Correspondence: chillxichen@google.com

train a 4B-parameter model, which we call ViT-e ("enormous"). ViT-e achieves good performance on image-only tasks, such as 90.9% ImageNet fine-tuning, and 84.9% on ObjectNet (Barbu et al., 2019).

We find benefits from jointly scaling both the vision and the language components, with vision providing a better return on investment (accuracy improvement per parameter/FLOP). As a result, the capacity of our largest PaLI model, PaLI-17B, is distributed relatively equitably between the two modalities, with the ViT-e component accounting for about 25% of the total parameter count. This is not always the case for prior work in large-capacity vision and language modeling (Wang et al., 2022a; Alayrac et al., 2022), due to the prior scale mismatch between vision and language backbones. We enable knowledge-sharing between multiple image and/or language tasks by casting them into a generalized VQA-like task. We frame all tasks using an "image+query to answer" modeling interface, in which both the query and answer are expressed as text tokens. This allows PaLI to capitalize on transfer learning across tasks, and enhance language-and-image understanding capabilities in a wide range of vision and language problems: image captioning, visual question-answering, scene-text understanding, and others (Figure 1).

To train PaLI-17B, we build a new high-volume image-and-language dataset WebLI, which consists of 10 billion images and tens of billions of image-text pairs. Importantly, the WebLI dataset contains text in over 100 languages. By training the model to perform multimodal tasks in many languages, we greatly increase the task diversity, and test the model's ability to effectively scale both across tasks and across languages. As a reference for future usage, we provide a data card to report information about the WebLI and its construction.

PaLI-17B achieves state-of-the-art (SOTA) results on multiple benchmarks, outperforming some strong models. Specifically, PaLI outperforms recent and concurrent models on the long-standing COCO Captioning benchmark (Chen et al., 2015), with **149.1** CIDEr score on the Karpathy split (Karpathy & Fei-Fei, 2015). PaLI also achieves a new SOTA of **84.3%** on VQAv2 (Goyal et al., 2017) while using an open-vocabulary text generative setting that is similar to Flamingo (Alayrac et al., 2022). This result outperforms even models evaluated in a fixed-vocabulary classification setting, e.g. CoCa (Yu et al., 2022), SimVLM (Wang et al., 2021), BEiT-3 (Wang et al., 2022c). Last but not least, our work provides a scaling roadmap for future multimodal models. Our results support the conclusion that scaling the components of each modality yields better performance compared to more skewed alternatives. Model scaling is also important for language-image understanding in multiple languages. In summary, our contributions are the following:

- We design a simple, modularized and scalable sequence-to-sequence learning architecture that can be efficiently trained by reusing existing Transformer-based unimodal checkpoints.

- We perform joint scaling on both the language and vision components for a wide range of parameters, and show no saturation of performance on both components for the largest model size we consider, PaLI-17B. More importantly, we show that multimodal performance greatly benefits from scaling the vision component beyond the previous-largest ViT, which provides a scaling roadmap for future vision & language models.

- We empirically validate that a mixture-of-objectives benefits the performance of large vision & language models.

- We scale up pre-training data to include over 100 languages, and train a large-capacity multilingual multimodal model. We show that a properly-scaled model can handle well a large number of languages, while still achieving SOTA performance on English-only tasks.

## 2 RELATED WORK

Pre-trained models have proven effective in both vision (Dosovitskiy et al., 2021; Zhai et al., 2022a) and language (Raffel et al., 2020; Brown et al., 2020) tasks. Image-text pre-training has also become the default approach to tackle V&L tasks (Tan & Bansal, 2019; Chen et al., 2020; Zhang et al., 2021; Cho et al., 2021; Hu et al., 2022). While benefiting from the text representation and generation capabilities of the Transformer architecture, some of these vision-language models rely on external systems (such as Fast(er) R-CNN (Ren et al., 2015)) to provide detected object names and the related precomputed dense features. Such reliance limited the capability to scale up the model and performance. With the introduction of Vision Transformers (Dosovitskiy et al., 2021), vision and

language modalities can be jointly modeled by transformers in a more scalable fashion (Yuan et al., 2021; Yu et al., 2022; Wang et al., 2022a; Alayrac et al., 2022).

One approach for image-text pre-training is contrastive learning (Radford et al., 2021; Jia et al., 2021). Zhai et al. (2022b) show that with a pre-trained and locked vision model, one needs to train only a paired text encoder model to get good language embeddings. Yuan et al. (2021) extend contrastively pre-trained models to more downstream tasks with task-specific adaptations. Beside image and language, MERLOT (Zellers et al., 2021) has found success in video understanding and reasoning through video-language pretraining. Another approach is to train vision-language models to generate text autoregressively (Donahue et al., 2015; Vinyals et al., 2015). This approach has the advantage of a unified formulation of vision-language tasks as a text generation problem (Cho et al., 2021; Wang et al., 2022b; Piergiovanni et al., 2022b). In Cho et al. (2021), the vision-language model is trained to recover masked text. SimVLM (Wang et al., 2021) propose an image-language pre-training approach leveraging a prefix language modeling objective. The unified framework OFA (Wang et al., 2022b) extends the generation capability to include text to image generation. Concurrent with our work, Unified-IO (Lu et al., 2022) further scaled up the number of objectives and tasks and demonstrated decent performance across the board through only multi-task pre-training without task-specific fine-tuning.

Recent works explore joint vision and language modeling with increased model capacity. CoCa (Yu et al., 2022) pre-trains a 2.1B image-text encoder-decoder model jointly with contrastive loss and generative loss. GIT (Wang et al., 2022a) trains a model consisting of a single image encoder and a text decoder with a captioning (generative) loss, where the image encoder is pre-trained with contrastive loss. In their latest version, GIT2, the model size is scaled up to 5.1B, with the majority of parameters on the vision side (4.8B). BEiT-3 (Wang et al., 2022c) presents an architecture with vision, language, and vision-language experts, operating with a shared multi-head self-attention followed by a switch for "expert" modules, resulting in a 1.9B model trained from scratch on a variety of public image, text and image-text datasets. Flamingo (Alayrac et al., 2022) is built upon a 70B language model (Hoffmann et al., 2022) as a decoder-only model whose majority of parameters are frozen in order to preserve language-generation capabilities, along with a 435M vision encoder.

Vision-language pre-training also benefits from automatically mined and filtered large-scale datasets such as Conceptual Captions (CC3M) and CC12M (Sharma et al., 2018; Changpinyo et al., 2021), with 3 and 12 million image-text pairs, respectively. With more relaxed filtering, LEMON (Hu et al., 2022) collected a larger dataset with 200M examples, which is further expanded to 800M examples in GIT (Wang et al., 2022a). For better scaling the model, larger, noisier datasets such as the ALIGN dataset (1.8B) (Jia et al., 2021) have been constructed, which has benefited SimVLM (Wang et al., 2021) and CoCa (Yu et al., 2022). While these image-text datasets have fueled the foundational V&L models with state-of-the-art performance, they are English-only, and there has been limited attempts to create datasets not English-centric and unlock the multilingual capability of these models.

## 3 THE PALI MODEL

### 3.1 ARCHITECTURE

With PaLI, we aim to perform both unimodal (language, vision) and multimodal (language and vision) tasks. Typically, many of these tasks are best handled by different models. For instance, image classification, and many formulations of VQA, require predicting elements from a fixed set, while language-only tasks and image captioning require open-vocabulary text generation. Similar to the recent work OFA (Wang et al., 2022b) and a concurrent work (Lu et al., 2022), we resolve this by using a sufficiently general interface for all tasks considered: the model accepts as input an image and text string, and generates text as output. The same interface is used both during pre-training and fine-tuning. Since all tasks are performed with the same model, i.e. we have no tasks-specific parameters or "heads", we use text-based prompts to indicate to the model which task to perform.

Figure 1 shows a high-level schematic of the model architecture. At its core, PaLI has a text encoder-decoder Transformer (Vaswani et al., 2017). To include vision as input, the text encoder is fed with a sequence of visual "tokens": output patch features of a Vision Transformer which takes as input an image. No pooling is applied to the output of the Vision Transformer before passing the visual tokens to the encoder-decoder model via cross-attention. We reuse previously trained unimodal checkpoints.

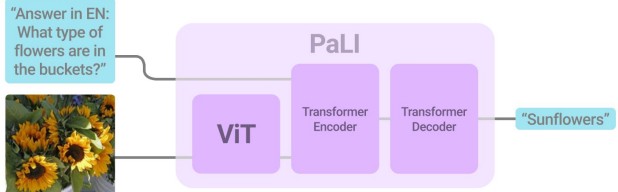

Figure 1: The PaLI main architecture is simple and scalable. It uses an encoder-decoder Transformer model, with a large-capacity ViT component for image processing.

For the text encoder-decoder, we reuse pre-trained mT5 (Xue et al., 2021) models, while for the image encoder, we reuse large vanilla ViT models (Dosovitskiy et al., 2021; Zhai et al., 2022a).

**The visual component** We introduce and train the largest vanilla ViT architecture to date, named **ViT-e**. ViT-e has the same architecture and uses the same training recipe as the 1.8B parameter ViT-G model (Zhai et al., 2022a), while scaling to 4B parameters. The only other difference is that we apply learning rate cool-down twice, once with and once without inception crop augmentation, and average ("soup") the weights of the two models as in Wortsman et al. (2022). While the scaling laws have been studied in both the vision domain and the language domain, scaling behaviour is less explored in combined vision and language models. Scaling up vision backbones leads to saturating gains on classification tasks such as ImageNet (Zhai et al., 2022a). We further confirm this, observing that ViT-e is only marginally better than ViT-G on ImageNet (Table 16). However, we observe substantial performance improvements from ViT-e on vision-language tasks in PaLI (Section 4). For example, ViT-e yields almost three additional CIDEr points over ViT-G on the COCO captioning task. This hints towards future headroom for vision-language tasks with even larger ViT backbones.

**The language component** We adopt the mT5 (Xue et al., 2021) backbone as our language component. We experiment using the pre-trained mT5-Large (1B parameters) and mT5-XXL (13B parameters), from which we initialize the language encoder-decoder of PaLI. We train on a mix of many tasks, including pure language understanding tasks (Section A.2). This helps avoid catastrophic forgetting of the mT5's language understanding and generation abilities. As a result, PaLI-17B continues to achieve similar levels of language-understanding accuracy on both the English benchmarks (Wang et al., 2019a) and across languages measured by the XTREME benchmark (Hu et al., 2020) (Section 4).

**The overall model** Three model sizes are considered (Table 7): 1) PaLI-3B, where the language component is initialized from mT5-Large (Xue et al., 2021) (1B parameters), and the vision component is ViT-G (Zhai et al., 2022a) (1.8B parameters). 2) PaLI-15B, where the language component is initialized from mT5-XXL (Xue et al., 2021) (13B parameters), and the vision component is ViT-G (1.8B parameters). 3) PaLI-17B, where the language model is initialized from mT5-XXL, and the vision component is the newly-trained ViT-e model (4B parameters).

## 3.2 DATA

**WebLI Dataset** Scaling studies for deep learning show that larger models require larger datasets to train effectively (Hoffmann et al., 2022; Kaplan et al., 2020; Zhai et al., 2022a). To unlock the potential of multilingual image-language pre-training, we introduce WebLI, a multilingual image-language dataset built from images and texts available on the public web. WebLI scales up the image language data collection from English-only datasets to 109 languages, which enables us to pre-train PaLI multilingually, and perform downstream tasks across many languages. The data collection process is similar to those reported in (Jia et al., 2021; Zhai et al., 2022b). Due to the abundance of multilingual content on the internet, the collection process for the WebLI dataset can be scaled to cover 10 billion images and 12 billion alt-texts. In addition to annotation with web text, we use publicly available automatic service to extract OCR annotations on all images, resulting in 29 billion image-OCR pairs. To balance quality and retain scale, we filter the dataset to the highest quality subset retaining only the top 10% scoring of the original WebLI image-text pairs (about 1B examples), which we use to train PaLI. Examples and statistics for the WebLI corpus and a complete datasheet (Pushkarna et al., 2022) are shown in Appendix B (Figure 4) and G.

**Training mixture** To accommodate diverse tasks in the image-language space, we train PaLI using a mixture of eight pre-training tasks. This mixture is designed to span a range of general capabilities useful for downstream tasks. **Span corruption on text-only data** uses the same technique described by Xue et al. (2021) on text-only examples. **Split-captioning on WebLI alt-text data** is inspired by the pre-training objective of Wang et al. (2021), and works by splitting each alt-text string randomly into two parts, $\langle \mathrm{cap}_1 \rangle$ and $\langle \mathrm{cap}_2 \rangle$, used for input and target, respectively. **Captioning on CC3M-35L** with the alt-text string in language $\langle \mathrm{lang} \rangle$ as the target, based on the Conceptual Captions (Sharma et al., 2018) training data and machine translated alt-texts. **OCR on WebLI OCR-text data** uses the concatenation of the annotated OCR texts in language $\langle \mathrm{lang} \rangle$ (Kil et al., 2022) produced by publicly available automatic service for the input image. **English and Cross-Lingual VQA** is VQ$^2$A-CC3M (Changpinyo et al., 2022a), translated in the same way as CC3M-35L. Note that we use English answers in all instances here, as the English-native answers for VQA are often short and too prone to errors to perform out-of-context automatic translation. **English and Cross-Lingual visual question generation (VQG)** is also based on native and translated VQ$^2$A-CC3M-35L VQA triplets. Similarly, we use only English answers here. **English-only Object-Aware (OA) VQA** is based on VQA triplets derived from automatically-produced, non-exhaustive object labels, inspired by Piergiovanni et al. (2022a). The QA pairs include listing all the objects in the image and whether a subset of objects are in the image. To create these examples, we require object-level annotations, for which we use Open Images (Kuznetsova et al., 2020). **Object detection** is a generative object-detection task inspired by Chen et al. (2021; 2022).

We specify each task using a training data source and a template-based prompt, and train the model using a language-model–style teacher forcing (Goodfellow et al., 2016) with a standard softmax cross-entropy loss. The coefficients for the training mixture are empirically determined, with 1.6B total examples in the mixture (Appendix A.2). The whole mixture is slightly smaller and designed to be cleaner than the datasets used in SimVLM (1.8B), CoCa (1.8B), and Flamingo (2.3B). However, unlike the aforementioned datasets, examples in our 1.6B dataset follow a long-tailed distribution over the 100+ languages covered. To prevent leakage between the pre-training examples and the downstream benchmarks. WebLI has undergone near de-duplication (Jia et al., 2021) of the images against the train, validation, and test splits of 68 common vision/vision-language datasets. For other datasets in the mixture, we performed the same de-duplication against all the downstream tasks.

### 3.3 Model Training

All PaLI variants are trained for one epoch over the entire pre-training dataset (1.6B) with $224 \times 224$ image resolution. Only the parameters of the language component are updated, the vision component is frozen, which is beneficial (Sec. 4.6). For the largest model, PaLI-17B, we perform an additional high-res ($588 \times 588$) phase similar to previous works (Radford et al., 2021; Yuan et al., 2021; Yu et al., 2022). This phase is only for 10k steps, covering 10M examples in total, with all the parameters of PaLI updated. More details for training PaLI and the ViT-e backbone are in Appendix A.1.

## 4 Experiments

We fine-tune and evaluate PaLI-3B and PaLI-15B checkpoints at $490 \times 490$ resolutions. For PaLI-17B, unless otherwise stated, the checkpoint produced by the two-phase pre-training is fine-tuned and evaluated at $588 \times 588$ resolution. For all the benchmarks, cross-entropy loss is used for fine-tuning.

### 4.1 Image Captioning

We fine-tune on **COCO Captions** (Chen et al., 2015) on the widely adopted Karpathy split (Karpathy & Fei-Fei, 2015). PaLI outperforms the latest SOTA trained with cross-entropy loss (Wang et al., 2022c), and establishes a new high of CIDEr score (Vedantam et al., 2015) at 149.1 (Table 1) for models without CIDEr-optimization. **NoCaps** (Agrawal et al., 2019) is an evaluation benchmark for image captioning that has similar style to COCO, but targets many more visual concepts than those included in the COCO. We follow previous works by evaluating NoCaps using a model fine-tuned on COCO. PaLI-17B achieves a 124.4 CIDEr score on test, comparable to the recent result of 124.8 from GIT2 (Wang et al., 2022a). GIT2 achieves 124.2, 125.5, 122.3 on in-domain, near-domain, and out-of-domain splits of the NoCaps test set, respectively. PaLI-17B achieves 121.1,

124.4 and 126.7, respectively. This suggests that for PaLI-17B, the domain transfer from COCO to NoCaps is slightly sub-optimal compared with models pre-trained with English only. Nevertheless, PaLI-17B outperforms all prior models on recognizing and describing long-tail objects outside of COCO's domain. **TextCaps** (Sidorov et al., 2020) focuses on captioning for images containing text. **VizWiz-Cap** (Gurari et al., 2020) contains images taken by people who are blind, which also involves scene-text understanding. We fine-tune on TextCaps and VizWiz-Cap using OCR strings generated by publicly available automatic service, similar to the protocol used in (Yang et al., 2021). Further details, including results evaluating PaLI-17B without OCR as input, are provided in Appendix C.5.

Table 1: CIDEr results for image captioning over the English benchmarks COCO Captions (Karpathy split), NoCaps, TextCaps, and VizWiz-Cap.

| Model | COCO Karpathy-test | NoCaps val | test | TextCaps val | test | VizWiz-Cap test-dev | test-std |
|---|---|---|---|---|---|---|---|
| LEMON (0.7B) | 139.1 | 117.3 | 114.3 | - | - | - | - |
| SimVLM | 143.3 | 112.2 | 110.3 | - | - | - | - |
| CoCa (2.1B) | 143.6 | 122.4 | 120.6 | - | - | - | - |
| GIT (0.7B) | 144.8 | 125.5 | 123.4 | 143.7 | 138.2 | 113.1 | 114.4 |
| GIT2 (5.1B) | 145.0 | **126.9** | **124.8** | 148.6 | 145.0 | 119.4 | 120.8 |
| OFA (0.9B) | 145.3 | - | - | - | - | - | - |
| Flamingo (80B) | 138.1 | - | - | - | - | - | - |
| BEiT-3 (1.9B) | 147.6 | - | - | - | - | - | - |
| PaLI-3B | 145.4 | 121.1 | - | 143.6 | - | 117.2 | - |
| PaLI-15B | 146.2 | 121.2 | - | 150.1 | - | 121.7 | - |
| PaLI-17B | **149.1** | **127.0** | 124.4 | **160.0** | **160.4** | **123.0** | **124.7** |

**Multilingual captioning on Crossmodal-3600** Following Thapliyal et al. (2022), we fine-tune PaLI models on COCO-35L, which is COCO captions translated into 35 languages similar to CC3M-35L, before evaluating on Crossmodal-3600. We used the checkpoints pre-trained at $224 \times 224$ resolution and fine-tuned on COCO-35L at the same resolution. We normalize the unicode, tokenize, and remove all punctuation before calculating CIDEr scores. For languages without word boundaries such as Chinese, Japanese, Korean and Thai, a neural model is used for segmenting the text. To illustrate the range of improvements over a variety of language families with different scripts and different resources, we use seven languages in Table 2 to show their exact CIDEr scores, in addition to the 35-language average score. PaLI outperforms previous SOTA by large margins. Note that due to different linguistic structures, the variance of CIDEr scores across different languages does not indicate lower quality of prediction on certain languages. In Appendix C.2, we back-translate the non-English predictions to English, and demonstrated that the capability of PaLI on both English and other languages is rather consistent.

Table 2: CIDEr scores on image captioning for the Crossmodal-3600 benchmark for seven diverse languages (English, French, Hindi, Hebrew, Romanian, Thai, and Chinese), as well as the average of the 35 languages covered by the benchmark.

| Model | en | fr | hi | iw | ro | th | zh | 35-lang avg. |
|---|---|---|---|---|---|---|---|---|
| Thapliyal et al. (2022) (0.8B) | 57.6 | 40.9 | 20.6 | 16.1 | 13.9 | 35.5 | 19.8 | 28.9 |
| PaLI-3B | 92.8 | 68.6 | 30.3 | 39.2 | 30.3 | 65.9 | 32.2 | 47.0 |
| PaLI-17B | **98.1** | **75.5** | **31.3** | **46.8** | **35.8** | **72.1** | **36.5** | **53.6** |

## 4.2 VISUAL QUESTION ANSWERING

All the VQA fine-tuning experiments in this paper are performed in the open-vocabulary setting using the 250k mT5 (Xue et al., 2021) vocabulary (Table 3). Most prior works, e.g. SimVLM (Wang et al., 2021), CoCa (Yu et al., 2022) and BEiT-3 (Wang et al., 2022c), use the VQA-as-classification setting, where the best answer among a predefined set (usually of size 3k) needs to be selected. Note that the VQA-as-open-generation setting is challenging because: (1) The generated text is directly compared to the desired answer and only an exact match is counted as accurate. (2) The PaLI vocabulary covers 100+ languages and is significantly larger than both those used in the classification setting, and those used by previous single-language open-generation models (Alayrac et al., 2022; Wang et al., 2022a).

Table 3: VQA Accuracy results on VQAv2, OKVQA, TextVQA, VizWiz-QA, and ANLS result on ST-VQA. PaLI models are evaluated in the open-vocabulary generation setting, and still outperform previous models that use closed-vocabulary classification (SimVLM, CoCa, BEiT-3, OFA). The result on OKVQA by Flamingo (with "*") is obtained in a 32-shot learning setup. Mia (Qiao et al., 2021) (with "†") is the winning model of TextVQA Challenge 2021, based on fine-tuning T5-XL (Raffel et al., 2020). Numbers shown in gray are from models using closed-vocabulary classification.

| | VQAv2 | | OKVQA | TextVQA | | VizWiz-QA | | ST-VQA | |
| Method | test-dev | test-std | val | val | test | test-dev | test | val | test |
|---|---|---|---|---|---|---|---|---|---|
| SimVLM | 80.03 | 80.34 | - | - | - | - | - | - | - |
| CoCa (2.1B) | 82.3 | 82.3 | - | - | - | - | - | - | - |
| GIT (0.7B) | 78.56 | 78.81 | - | 59.93 | 59.75 | 68.0 | 67.5 | 69.1 | 69.6 |
| GIT2 (5.1B) | 81.74 | 81.92 | - | 68.38 | 67.27 | 70.97 | 70.1 | 75.1 | 75.8 |
| OFA (0.9B) | 82.0 | 82.0 | - | - | - | - | - | - | - |
| Flamingo (80B) | 82.0 | 82.1 | 57.8* | 57.1 | 54.1 | 65.7 | 65.4 | - | - |
| BEiT-3 (1.9B) | 84.2 | 84.0 | - | - | - | - | - | - | - |
| KAT | - | - | 54.4 | - | - | - | - | - | - |
| Mia | - | - | - | - | 73.67† | - | - | - | - |
| PaLI-3B | 81.4 | - | 52.4 | 60.12 | - | 67.5 | - | 67.5 | 69.7 |
| PaLI-15B | 82.9 | - | 56.5 | 65.49 | - | 71.1 | - | 73.2 | 76.5 |
| PaLI-17B | **84.3** | **84.3** | **64.5** | **71.81** | 73.06 | **74.4** | **73.3** | **77.1** | **79.9** |

On **VQAv2**, PaLI achieves 84.3 accuracy on VQAv2, and outperforms previous SOTA as follows: (1) By +2.2 accuracy points on the open-vocabulary generation setting, compared to Flamingo (Alayrac et al., 2022). (2) By +0.3 accuracy points when compared against the best result on the closed-vocabulary classification setting, BEiT-3 (Wang et al., 2022c). **OKVQA** requires external knowledge to answer its questions, that is, knowledge not directly present in the image input, and instead needs to be indirectly inferred by the model. PaLI-17B achieves 64.5 accuracy, pushing SOTA for the pretrain-finetune setup higher by 10.1 accuracy points, compared to KAT (Gui et al., 2021) at 54.4 accuracy. The best result for the 32-shot learning setup is from Flamingo (Alayrac et al., 2022) at 57.8 accuracy. The results from Flamingo and PaLI-17B suggest that leveraging external knowledge does not necessarily require specific training, and instead can be achieved with generic large-capacity models trained on large amounts of data. **TextVQA** (Singh et al., 2019), **VizWiz-QA** (Gurari et al., 2018) and **ST-VQA** (Biten et al., 2019) require the ability to perform question answering in the presence of text in the input image. We fine-tune using OCR strings generated by publicly available automatic service, similar to the protocol in TAP (Yang et al., 2021) and Mia (Qiao et al., 2021). Evaluation on TextVQA and VizWiz-QA without OCR as input is provided in Appendix C.5.

**Cross-lingual and Multilingual VQA on xGQA and MaXM** Both xGQA (Pfeiffer et al., 2022) and MaXM (Changpinyo et al., 2022b) are test-only VQA benchmarks that require multilingual understanding of visual questions. The setting in xGQA is cross-lingual (English-answers only), whereas for MaXM it is multilingual (answer in the same language as the question). We evaluate PaLI-17B pre-trained at 224 image resolution and fine-tuned on the native and translated VQAv2 (Goyal et al., 2017) (the Karpathy train split) in the 13 languages covered by xGQA and MaXM (VQAv2-13L) at 378 resolution. Table 4 shows significant gains on both benchmarks across all languages.

Table 4: Cross-lingual VQA results on xGQA (Pfeiffer et al., 2022) (left) and multilingual VQA results on MaXM (Changpinyo et al., 2022b) (right). All models are fine-tuned on translated VQAv2 in 13 languages. Exact-match accuracy is reported. Referenced MPT results are from (Changpinyo et al., 2022b)

| | xGQA | | | | | | | | MaXM | | | | | | |
| Model | en | bn | de | id | ko | pt | ru | zh | en | fr | hi | iw | ro | th | zh |
|---|---|---|---|---|---|---|---|---|---|---|---|---|---|---|---|
| MPT | 41.5 | 38.6 | 40.5 | 39.5 | 38.7 | 39.8 | 39.5 | 39.5 | 36.6 | 36.2 | 55.1 | 40.6 | 42.3 | 50.0 | 30.3 |
| PaLI-17B | **54.2** | **50.0** | **52.2** | **50.6** | **50.4** | **51.3** | **50.3** | **50.6** | **56.4** | **46.4** | **67.3** | **60.0** | **57.4** | **65.6** | **46.9** |

## 4.3 LANGUAGE-UNDERSTANDING CAPABILITIES

Since PaLI is pre-trained with a diverse mixture of multimodal tasks with image and text data, it raises the question on whether it would "forget" its language modeling capability, and therefore

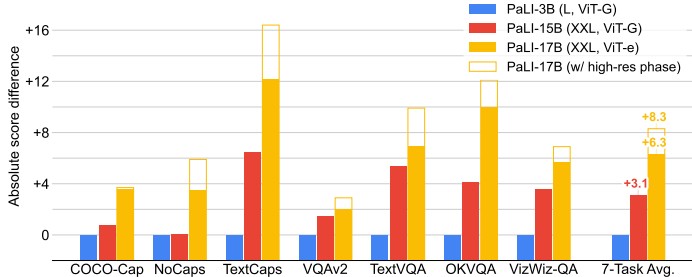

Figure 2: PaLI scaling for a number of tasks. We report CIDEr scores for captioning tasks, and accuracy scores for VQA tasks. Both scaling the language side (from 1B to 13B parameters) and the vision side of the model (from 2B to 4B parameters) yield improvements across all tasks. The results represented by solid bars are from the standard 224×224 resolution pre-training. The empty orange bars correspond to PaLI-17B checkpoints with the high resolution pre-training phase.

exhibit inferior performance on language-understanding tasks compared to its unimodal starting checkpoint (mT5-XXL in the case of PaLI-17B). Therefore, we compare mT5-XXL and PaLI-17B on a range of language understanding benchmarks, including the English-only SuperGLUE benchmark (Wang et al., 2019a), as well as three multilingual benchmarks from the XTREME (Hu et al., 2020): XNLI (Conneau et al., 2018), which is a textual entailment task covering 14 languages, XQuAD (Artetxe et al., 2020) and TyDiQA-GoldP (Clark et al., 2020), which are both question-answering tasks covering 10 and 11 languages, respectively. For the three XTREME benchmarks, we evaluate in the zero-shot (ZS) transfer setting, whereas for SuperGLUE the models are fine-tuned (FT). Table 11 in Appendix C.1 summarizes the results. Despite the pre-training mixture heavily favoring the V&L tasks, PaLI-17B is able to maintain a high-level of language-understanding capabilities for English, and it is on-par with the state-of-the-art mT5-XXL checkpoint on the XTREME benchmarks.

## 4.4 ZERO-SHOT IMAGE CLASSIFICATION

We evaluate the PaLI checkpoints (without high-res phase) at 224×224 resolution on ImageNet and ImageNet OOD evaluation sets: ImageNet (Deng et al., 2009), ImageNet-R (Hendrycks et al., 2021a), ImageNet-A (Hendrycks et al., 2021b), ImageNet-Sketch (Wang et al., 2019b), ImageNet-v2 (Recht et al., 2019) and ObjectNet (Barbu et al., 2019). We use the same interface as for all other tasks. Instead of training a classifier on top of PaLI, we condition on the image and use PaLI's decoder to score strings corresponding to each class directly. (See Appendix C.8 for details) The top-1 accuracies are presented in Table 5, where it clearly shows that PaLI-17B is significantly better than smaller variants. We are not aware of any previous work for large scale zero-shot evaluation on ImageNet with a generative model. However, PaLI with a zero-shot setting outperforms the 1-shot learning result from Flamingo (Alayrac et al., 2022).

Table 5: Top 1 accuracy results of 0-shot image classification on ImageNet, ImageNet-R, ImageNet-A, ImageNet-Sketch, Imagenet-v2, and ObjectNet. Top-5 results are in the Appendix (Table 21).

| Model (ImageNet data) | INet | INet-R | INet-A | INet-Sketch | INet-v2 | ObjNet |
|---|---|---|---|---|---|---|
| Flamingo-80B (**1-shot**) | 71.9 | - | - | - | - | - |
| Flamingo-80B (**5-shot**) | 77.3 | - | - | - | - | - |
| PaLI-3B (**0-shot**) | 70.06 | 80.15 | 37.92 | 61.11 | 62.55 | 38.87 |
| PaLI-15B (**0-shot**) | 70.27 | 81.21 | 41.16 | 61.03 | 62.81 | 39.51 |
| PaLI-17B (**0-shot**) | **72.11** | **81.97** | **44.70** | **63.83** | **64.46** | **42.62** |

## 4.5 MODEL SCALING

Due to the modular architecture, the image and language components of PaLI can be scaled independently. We demonstrate that jointly scaling the capacity of both components leads to performance improvements. Figure 2 quantifies this improvement across seven V&L benchmarks where we

have also evaluated the PaLI-17B checkpoint without the high resolution pre-training phase for fair comparison. These improvements are noticeable both when scaling the language-model capacity (from L to XXL), and the vision-model capacity (from ViT-G to ViT-e). Figure 2 also shows that scaling the visual component is important: when scaling from a ViT-G to a ViT-e model, although the overall model size is increased by only about 13% (+2B parameters), the average performance improvement over all seven benchmarks (additional +3.2) is larger than the one obtained with much larger increases in the capacity of the language model (+3.1) which takes more parameters (+12B). The high-resolution pre-training phase at 588×588 resolution brings an additional +2.0 points, which also indicates the potential of scaling up the vision component of the model. This observation also resonates with the significant improvement from PaLI-15B to 17B on generative ImageNet zero-shot classification (Table 5). Table 12 shows the results of a 5B version of PaLI with mT5-L and ViT-e on two benchmarks, which also resonates with the finding of the benefit of joint scaling. For context, in prior work, V&L scaling is usually conducted at lower model capacity: for instance, CoCa (Yu et al., 2022) scales up to 2.1B parameters, or scaling is done primarily via the language-modeling backbone, e.g. Flamingo (Alayrac et al., 2022) scales the text backbone to 80B but the image backbone remains at 435M. Finally, on the Crossmodal-3600 benchmark, we show that scale has a large impact on multilingual performance as well (Figure 5 in the Appendix).

## 4.6 ABLATION STUDIES

We examine the composition of the task mixture and demonstrate the effectiveness of our multiple-objective mixture design. To this end, we pre-train a PaLI-3B model with 200M data coverage for each setting, before fine-tuning on a combination of English and multilingual V&L tasks (Table 6). Aside from the four tasks from our main evaluation for PaLI, we also add a VQAv2-based VQG benchmark (Akula et al., 2021). The relative weight of each components remains the same as the full mixture (Table 9). As a first observation, the split-cap objective on WebLI appears to be the most critical, across all benchmarks. Second, the object-related components also boost performance on all benchmarks. Third, the captioning objective on CC3M-35L helps on COCO; on XM-3600, its positive contribution for non-EN languages and the slight degradation for English is a reflection of CC3M-35L having a much higher non-EN example ratio (34/35) compared to WebLI alt-text (60% English, Figure 4). Fourth, adding VQA helps TextVQA; in addition, the VQG objective improves the model's VQG capability without impacting the performance on other benchmarks. Last but not least, the OCR objective positively impacts OCR-related tasks such as TextVQA, at a slight negative impact on captioning performance. We also note that VQAv2, due to its large training set size, is much less sensitive to the change in pre-training mixture. In addition, we perform ablations to quantify the positive impact of initializing from uni-modal checkpoints, as opposed to from-scratch training (Table 13); the minor accuracy improvement from freezing the ViT backbone during pre-training (Table 14); the effect of pretraining with non-English WebLI examples on multi-(cross-)lingual performance (Table 15).

Table 6: Mixture of objectives (PaLI-3B). TextVQA is fine-tuned with 490×490 resolution, while all other benchmarks are fine-tuned with 224×224. Results for VQAv2 are on the Karpathy validation set. XM-3600 denotes Crossmodal-3600, and "6L" is the average of the six non-English languages in Table 2. The order in which the components are ablated follows the presented order in Sec. 3.2, and "object-related" refers to the object-aware QA and generative object detection components together. TextVQA is fine-tuned without detected OCR string to better showcase the model's OCR capability

| Component | COCO | TextVQA | VQAv2 | XM-3600 (EN / 6L) | VQG (ZS / FT) |
|---|---|---|---|---|---|
| Full mixture | 141.4 | 41.6 | 76.0 | 93.8 / 42.5 | 96.7 / 194.0 |
| *w/o* split-cap | 140.4 (-1.0) | 38.8 (-2.8) | 75.5 (-0.5) | 87.5 (-6.3) / 41.5 (-1.0) | 86.3 (-10.4) / 190.5 (-3.5) |
| *w/o* captioning | 140.5 (-0.9) | 41.2 (-0.4) | 75.9 (-0.1) | 94.9 (+1.1) / 39.9 (-2.6) | 101.3 (+4.6) / 193.3 (-0.7) |
| *w/o* OCR | 142.3 (+0.9) | 39.9 (-1.7) | 75.9 (-0.1) | 95.4 (+1.6) / 43.6 (+1.1) | 92.5 (-4.2) / 193.7 (-0.3) |
| *w/o* VQA | 140.9 (-0.5) | 40.0 (-1.6) | 75.9 (-0.1) | 93.9 (+0.1) / 42.7 (+0.2) | 94.1 (-2.6) / 193.2 (-0.8) |
| *w/o* VQG | 141.4 (+0.0) | 41.3 (-0.3) | 75.8 (-0.2) | 95.1 (+1.3) / 42.0 (-0.5) | 17.9 (-78.8) / 188.2 (-5.8) |
| *w/o* object-related | 140.9 (-0.5) | 40.2 (-1.4) | 75.4 (-0.6) | 90.9 (-2.9) / 41.8 (-0.7) | 81.7 (-15.0) / 189.1 (-4.9) |

ETHICS STATEMENT AND BROADER IMPACTS

Large models may have broader societal impact. While such models have demonstrated strong performance on public benchmarks, they might contain unknown biases or stereotypes, or propagate inaccurate or otherwise distorted information. While we have made efforts to measure some of these issues, such models need to be re-assessed carefully before being used for specific purposes. The dataset used for pre-training is automatically harvested, and filtering of the data is automatic. That process may leave undesirable images or text annotations, descriptions or concepts to be incorporated into the model. We have also attempted to train the model to operate in more than 100 languages, which we believe is an important step forward for image-language models. However, languages have various levels of data presence and coverage, so the language-generated text varies in quality depending on the language, and might contain inaccurate or undesirable outputs.

REPRODUCIBILITY STATEMENTS

Our model is based on open sourced components - ViT and mT5 (Dosovitskiy et al., 2021; Xue et al., 2021). Model architecture details for each component is in Section 3.1. The configuration of ViT-e when scaling is provided in Table 7 and Section A.1. We have provided training and fine-tuning details in Section 3.3 and in Section A in the Appendix. Data and model cards are also provided in the Appendix.

ACKNOWLEDGEMENTS

We would like to thank Erica Moreira, Victor Gomes, Tom Small, Sarah Laszlo, Kathy Meier-Hellstern, Susanna Ricco, Emily Denton, Bo Pang, Wei Li, Jihyung Kil, Tomer Levinboim, Julien Amelot, Zhenhai Zhu, Xiangning Chen, Liang Chen, Filip Pavetic, Daniel Keysers, Matthias Minderer, Josip Djolonga, Ibrahim Alabdulmohsin, Mostafa Dehghani, Yi Tay, Rich Lee, Austin Tarango, Elizabeth Adkison, James Cockerille, Eric Ni, Anna Davies, Maysam Moussalem, Jeremiah Harmsen, Claire Cui, Slav Petrov, Tania Bedrax-Weiss, Joelle Barral, Tom Duerig, Paul Natsev, Fernando Pereira, Jeff Dean, and Zoubin Ghahramani for helpful discussions, feedback, and support.

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

# A  PaLI MODEL ADDITIONAL INFORMATION

## A.1  PaLI MODEL DETAILS

Figure 3 visualizes some examples of PaLI on several tasks, such as image captioning, visual question answering, OCR-oriented captioning and question answering. Examples in multiple languages are shown as well.

Below, we show more specifics about the PaLI model and its components.

**Model variants**    Table 7 lists the main PaLI models used where the largest is PaLI-17B of 17B parameters.

| Model | Components | Image Encoder | Multimodal Encoder-Decoder | Total |
|-------|-----------|---------------|----------------------------|-------|
| PaLI-3B | ViT-G, mT5-L | 1.8B | 1.2B | 3.0B |
| PaLI-15B | ViT-G, mT5-XXL | 1.8B | 13B | 14.8B |
| PaLI-17B | ViT-e, mT5-XXL | 3.9B | 13B | 16.9B |

Table 7: The size in terms of number of parameters for the trained PaLI model versions.

**ViT-e Backbone**    We show ViT-e's configuration in Table 8 alongside ViT-g and ViT-G for reference. Width, depth and MLP dimensions are all further scaled up in ViT-e, resulting in a model with 4B parameters. The model training setup is copied from the ViT-G model (Zhai et al., 2022a), on the JFT-3B dataset (Zhai et al., 2022a), with $16,384$ batch size, $224 \times 224$ resolution. We train the model for 1M steps using 0.0008 initial learning rate, with an inverse square-root learning rate decay, and a linear cool-down to zero for the final 100k steps. The only additional technique added is model souping (Wortsman et al., 2022): we run the 900K to 1M cool-down twice, once with inception cropping and once with resizing only. Thus, the final ViT-e model consists of the average weights of these two cool-downs. ViT-e is pretrained using the `big_vision` codebase (Beyer et al., 2022).

| Name | Width | Depth | MLP | Heads | Params (M) | GFLOPs | |
|------|-------|-------|-----|-------|-----------|--------|--------|
| | | | | | | $224^2$ | $384^2$ |
| g/14 | 1408 | 40 | 6144 | 16 | 1011 | 533.1 | 1596.4 |
| G/14 | 1664 | 48 | 8192 | 16 | 1843 | 965.3 | 2859.9 |
| e/14 | 1792 | 56 | 15360 | 16 | 3926 | 1980 | 5777 |

Table 8: ViT-e architecture details.

**The overall model**    The overall PaLI models are implemented in `JAX/Flax` (Bradbury et al., 2018) using the open-source `T5X` (Roberts et al., 2022) and `Flaxformer` (Heek et al., 2020) frameworks. For the learning rate, we use a 1k-step linear warmup, followed by inverse square-root decay. For PaLI-3B, we use a peak learning rate of 1e-2. For larger models, PaLI-15B and PaLI-17B, we use a peak learning rate of 5e-3. We use the Adafactor (Shazeer & Stern, 2018) optimizer with $\beta_1 = 0$ and second-moment exponential decay set to 0.8.

The largest model, PaLI-17B, is pretrained using 1,024 GCP-TPUv4 chips for 7 days. It uses a four-way model partitioning (Roberts et al., 2022) and a batch size of 4,096. This is slightly less TPU resources than used to train other large vision and language models on TPUs. SimVLM used 2,048 GCP-TPUv3 for 5 days (Wang et al., 2021), while CoCa used 2,048 GCP-TPUv4 chips for 5 days (Yu et al., 2022). Flamingo used 1,536 GCP-TPUv4 chips for 15 days (Alayrac et al., 2022).

During training, the model passes over 1.6B images, one epoch over the entire pretraining dataset. The image resolution for this pass is $224 \times 224$. During training, only the parameters of the language component are updated and the vision component is frozen, which provides a boost in performance (Sec. 4.6).

**Continuation of pretraining at higher image resolution** For the largest model, PaLI-17B, we perform a further high-resolution ($588 \times 588$) pre-finetuning for the multilingual tasks. When scaling

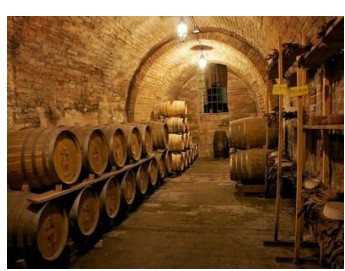

**Input**: Generate the alt_text in EN
**Output**: A cellar filled with barrels of wine

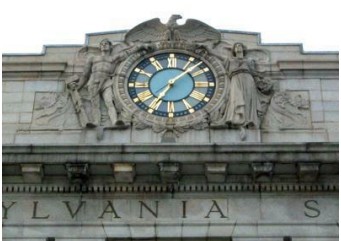

**Input**: Generate the alt_text in EN
**Output**: a clock on a building that says 'lyvania' on it

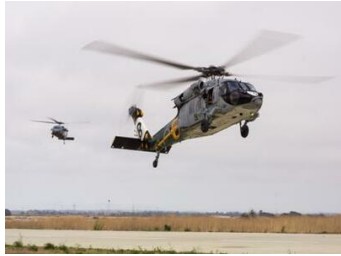

**Input**: Generate the alt_text in EN
**Output**: Two helicopters are flying in the sky and one has a yellow stripe on the tail

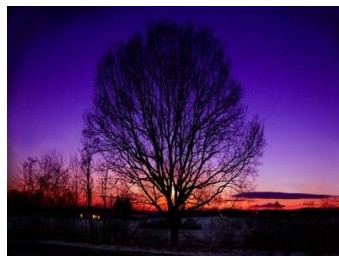

**Input**: Generate the alt_text in FR
**Output**: Un arbre debout dans un champ avec un ciel violet

*(A tree standing in a field with a purple sky)*

**Input**: Generate the alt_text in TH
**Output**: ลา สี เทา เดิน ไป ตาม ถนน

*(A gray donkey walks down the street)*

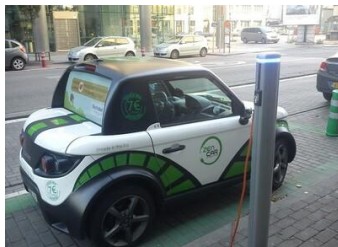

**Input**: Generate the alt_text in ZH
**Output**: 一辆 电动 汽车 停 在 充电 桩 上 。

*(An electric car parking on a charging station)*

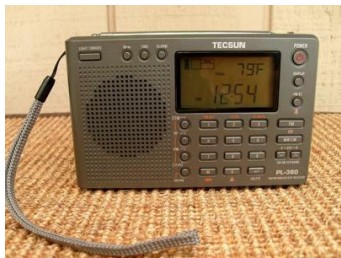

**Input**: Answer in EN: what time is it according to this radio
**Output**: 1254

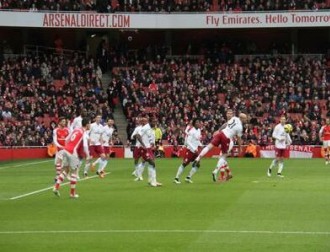

**Input**: Answer in EN: what website is on the wall in back
**Output**: arsenaldirect.com

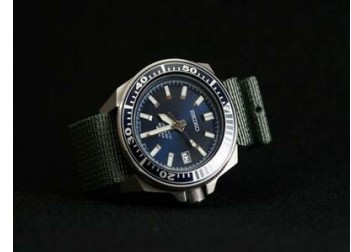

**Input**: Answer in EN: what is the brand of this watch
**Output**: seiko

Figure 3: PaLI addresses a variety of vision and language tasks across many languages, for example, image captioning, visual question answering, scene-text understanding, etc. Images from the publicly-available TextVQA (Singh et al., 2019) and TextCaps (Sidorov et al., 2020) datasets are shown, together with PaLI inputs and outputs.

up image resolution, the patch size is kept the same, and the number of patches are increased with higher resolution. We perform a 2D bilinear upsampling of the positional embedding to match the increased number of patches. This second stage of training is only for 10k steps at batch size 1024 (10M examples in total) and is performed on a subset of the full training mix. We simplify the mixture of data in this stage to focus on VQA, captioning and OCR capabilities, by including only the OCR, CC3M-35L and VQ$^2$A in the training mixture and making them equally weighted. In this high-resolution finetuning phase, all of the parameters of PaLI are updated. This high resolution phase was performed using 512 GCP-TPUv4 chips for an additional 3 days.

## A.2 THE PRETRAINING TASK MIXTURE

Below are detailed descriptions of each component of our task mixture.

- **Span corruption on text-only data** uses the same technique described by Xue et al. (2021), corrupting 15% of the tokens from a given text-only example and using "sentinels" of the form $\langle \text{extra\_id\_}k \rangle$ for each corrupted span; the text-only examples are using a sample of 100M of text-only examples.

- **Split-captioning (SplitCap) on WebLI alt-text data** is inspired by the pretraining objective of Wang et al. (2021), and works by splitting each alt-text string randomly into two parts, $\langle \text{cap}_1 \rangle$ and $\langle \text{cap}_2 \rangle$. It uses the prompt "*Generate the alt_text in* $\langle \text{lang} \rangle$ *at* $\langle \text{pos} \rangle$*:* $\langle \text{cap}_1 \rangle$ $\langle \text{extra\_id\_0} \rangle$" (where $\langle \text{lang} \rangle$ is the language code of the alt-text string, and $\langle \text{pos} \rangle$ is the number of words in $\langle \text{cap}_1 \rangle$), with $\langle \text{cap}_2 \rangle$ as the target.

- **Captioning (Cap) on CC3M-35L on native and translated alt-text data** using the prompt "*Generate the alt_text in* $\langle \text{lang} \rangle$ *at 0:* $\langle \text{extra\_id\_0} \rangle$", with the alt-text string in language $\langle \text{lang} \rangle$ as the target. CC3M-35L is Conceptual Captions (Sharma et al., 2018) training data, translated into an additional 34 languages (the same as the non-English ones covered by Crossmodal-3600 (Thapliyal et al., 2022), except for Cusco-Quechua), for a total of 100M examples.

- **OCR on WebLI OCR-text data** using the prompt "*Generate the ocr_text in* $\langle \text{lang} \rangle$*:* $\langle \text{extra\_id\_0} \rangle$", with $\langle \text{OCR\_text} \rangle$ as the target, where $\langle \text{OCR\_text} \rangle$ is the concatenation of the annotated OCR texts in language $\langle \text{lang} \rangle$ (Kil et al., 2022) produced by the publicly available automatic service for the input image.

- **English and Cross-Lingual VQA on native and translated** $\text{VQ}^2\text{A}$**-CC3M-35L-100M VQA triplets** using, for a given $\langle image, [question], [answer] \rangle$ VQA triple, the prompt: "*Answer in EN: [question]* $\langle \text{extra\_id\_0} \rangle$", with $[answer]$ for the target. $\text{VQ}^2\text{A}$-CC3M-35L-100M is a 100M random subset of $\text{VQ}^2\text{A}$-CC3M (Changpinyo et al., 2022a), translated into the same additional 34 languages as mentioned above. Note that we use English answers in all instances here, as the English-native answers for VQA are often short and too prone to errors to perform out-of-context automatic translation.

- **English and Cross-Lingual visual question generation (VQG) on native and translated** $\text{VQ}^2\text{A}$**-CC3M-35L-100M VQA triplets** using, for a given $\langle image, [question], [answer] \rangle$ VQA triple, the prompt: "*Generate a question in* $\langle \text{lang} \rangle$ *for [answer]:* $\langle \text{extra\_id\_0} \rangle$", with $[question]$ in language $\langle \text{lang} \rangle$ as the target. Similarly, we use only English answers here.

- **English-only Object-Aware (OA) VQA** is based on VQA triplets derived from automatically-produced, non-exhaustive object labels, inspired by Piergiovanni et al. (2022a). We automatically generate 4 different prompt types, based on the available object labels, as follows. (1) Prompt: "*Answer in EN: List the objects present:* $\langle \text{extra\_id\_0} \rangle$", with the target: $\langle \text{object}_1 \rangle, \ldots, \langle \text{object}_N \rangle$. (2) Prompt: "*Answer in EN: Is* $\langle \text{object}_k \rangle$ *in the image?* $\langle \text{extra\_id\_0} \rangle$", with the target "Yes" or "No". (3) Prompt: "*Answer in EN: Is* $\langle \text{object}_1 \rangle$, $\ldots, \langle \text{object}_N \rangle$ *in the image?* $\langle \text{extra\_id\_0} \rangle$", with the target "Yes" or "No". (4) Prompt: "*Answer in EN: Which of* $\langle \text{object}_1 \rangle, \ldots, \langle \text{object}_N \rangle$ *are in the image?* $\langle \text{extra\_id\_0} \rangle$", with the target made of the list of object labels present. To create these examples, we require object-level annotations, for which we use Open Images (Kuznetsova et al., 2020), from which we create 50M examples.

- **Object detection** is a generative object-detection task inspired by Chen et al. (2021; 2022). The target sequence describes bounding-box coordinates and object labels, e.g. "*10 20 90 100 cat 20 30 100 100 dog*". The coordinates are in the $y_{min} \, x_{min} \, y_{max} \, x_{max}$ order, and range between 0 and 999. Unlike Chen et al. (2021), the prompt used contains a set of positive and negative class labels, i.e. object classes that are present and not present in the image (e.g. "*detect cat and dog and leopard*"). The prompt is prefixed with the word "*detect*". For the datasets that do not have negative class labels explicitly defined, we randomly sample non-positive class labels. Since WebLI does not contain bounding box annotations, we train on a mixture of public datasets, totalling 16M images: Open Images (Kuznetsova et al., 2020), Visual Genome (Krishna et al., 2017), and Object365 (Shao et al., 2019). The datasets are de-duplicated against evaluation tasks. These examples are included to increase object awareness capabilities of the model.

**Dataset mixing ratio for pretraining** Table 9 provides the data mixing ratio for pretraining all PaLI variants.

| | Text-only | WebLI alt-text | OCR | CC3M-35L | VQA | VQG | OA | Detection | Total |
|---|---|---|---|---|---|---|---|---|---|
| Amount (M) | 100 | 1000 | 100 | 100 | 100 | 100 | 50 | 16 | 1566 |

Table 9: Mixing ratio of each task for pretraining

### A.3 FINE-TUNING DETAILS

**Hyperparameters for finetuning the V&L tasks** We performed limited hyperparameter search for finetuning. The train steps is mostly selected based on dataset size. The batch size is selected among {128, 256, 512}, and the initial learning rate among {1e-5, 3e-5, 1e-4}. The optimizer setting for finetuning is the same as the setting for pretraining. Note that we did not perform the hyperparameter sweep over all possible combinations. Table 10 summarizes the hyperparameters corresponding to the main results.

| Hyper-parameter | COCO & NoCaps | TextCaps | VizWiz-Cap | VQAv2 | TextVQA | VizWiz-QA | OKVQA | ST-VQA |
|---|---|---|---|---|---|---|---|---|
| Dropout | | | | 0.1 | | | | |
| LR decay schedule | | | | linear decay to zero | | | | |
| Train | 20k | 10k | 5k | 20k | 5k | 5k | 5k | 5k |
| Batch size | | | | 256 | | | | |
| Initial (peak) LR | 3e-5 | 1e-4 | 1e-4 | 1e-4 | 1e-4 | 1e-4 | 3e-5 | 1e-4 |

Table 10: Hyper-parameters used in fine-tuning experiments.

**Setup for zero-shot image classification** For each image, each class is scored using the prompt "*Generate alt_text in EN at 2: Photo of* $\langle$extra_id_0$\rangle$", scoring against all 1,000 classes with a target "$\langle en\_class\_name \rangle$", where "$\langle en\_class\_name \rangle$" stands for a classification label in English, such as *"goldfish"*, *"great white shark"*, etc.

## B WEBLI DATASET DETAILS

The WebLI dataset covers about 10 billion images and 12 billion alt-texts in 109 languages. We further apply a publicly available automatic service to extract OCR annotations on all images, producing additional 29 billion image-OCR pairs. Examples and statistics for the WebLI corpus are shown in Figure 4.

Due to the scale of WebLI, to mitigate train-to-test leakage, we perform near de-duplication of the images against the train, validation, and test splits of 68 common vision/vision-language datasets. Eliminating these images from the WebLI dataset does not result in any significant shrinkage (0.36%), and avoids any potential "leakage" of examples from the pretraining setup to the downstream evaluation tasks.

To improve the data quality in terms of image-text alignment, we score image and alt-text pairs based on their cross-modal similarity. This score is measured with cosine similarity between embedding representations from each modality, computed as follows. The image embeddings are trained with a graph-based, semi-supervised representation learning approach, as described in Juan et al. (2019). Then, the text embeddings are learned using the frozen image embeddings, based on a contrastive approach using a Transformer encoder for the text, which forces both modality representations to the same embedding space.

We tune a threshold on the image and alt-text pairs' score, and retain only the top 10% best scoring of the original WebLI image-text pairs (about 1B examples), which we use to train PaLI.

---

[1]The second image is by jopradier (original), used under the CC BY-NC-SA 2.0 license. Remaining images are also used with permissions.

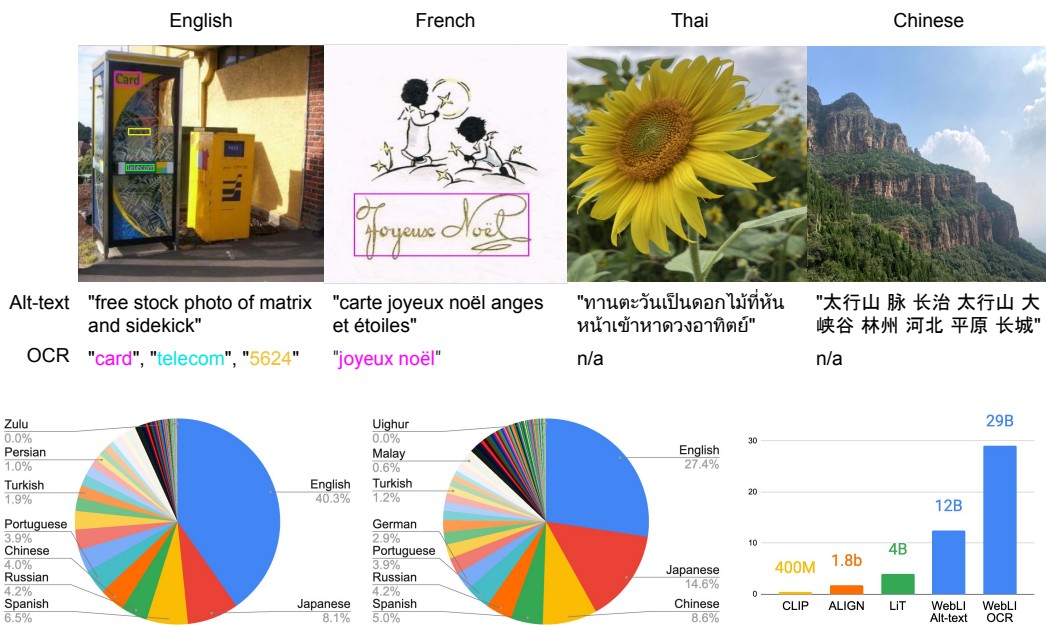

Figure 4: The WebLI dataset. Top: Sampled images[1] associated with multilingual alt-text (available) and OCR (computed using publicly available API ). Bottom left/middle: Statistics of recognized languages from alt-text/OCR. Bottom right: Image-text pair counts, compared against other large-scale vision-language datasets.

## C  ADDITIONAL EXPERIMENTAL RESULTS

### C.1  LANGUAGE-ONLY EVALUATION

In Table 11, we evaluate te performance of PaLI on a range of language understanding benchmarks, in order to verify that the language-only capabilities of the model have been preserved. More specifically we compare mT5-XXL and PaLI-17B, evaluating on the English-only SuperGLUE benchmark (Wang et al., 2019a), and on three multilingual benchmarks from the XTREME (Hu et al., 2020): XNLI (Conneau et al., 2018), which is a textual entailment task covering 14 languages, XQuAD (Artetxe et al., 2020) and TyDiQA-GoldP (Clark et al., 2020), which are both question-answering tasks covering 10 and 11 languages, respectively.

| Model
Method | SuperGLUE
FT | XNLI
ZS | XQuAD
ZS | TyDiQA-GoldP
ZS |
|---|---|---|---|---|
| Metric | Avg. Score | Accuracy | F1/EM | F1/EM |
| mT5-XXL (Xue et al., 2021) | 89.2 | 85.0 | 82.5 / 66.8 | 80.8 / 65.9 |
| mT5-XXL (our setting) | 89.3 | 84.5 | 82.6 / 66.6 | 81.6 / 66.3 |
| PaLI-17B | 88.2 | 84.9 | 81.8 / 66.0 | 81.2 / 66.5 |

Table 11: Results on SuperGLUE and three XTREME tasks. The first row is the result reported by mT5 (Xue et al., 2021) and ByT5 (Xue et al., 2022) paper. The second row is our repetition using the publicly available mT5-XXL checkpoint, which is also the starting point for PaLI-17B. The third row results are using the trained PaLI-17B model.

### C.2  ADDITIONAL SCALING RESULTS

Figure 5 shows that the model scaling impacts significantly the performance for multiple languages. We can see that PaLI-17B improves substantially over PaLI-3B across languages. We also include a plot where for a subset of 600 examples, we back-translate the predictions from six languages,

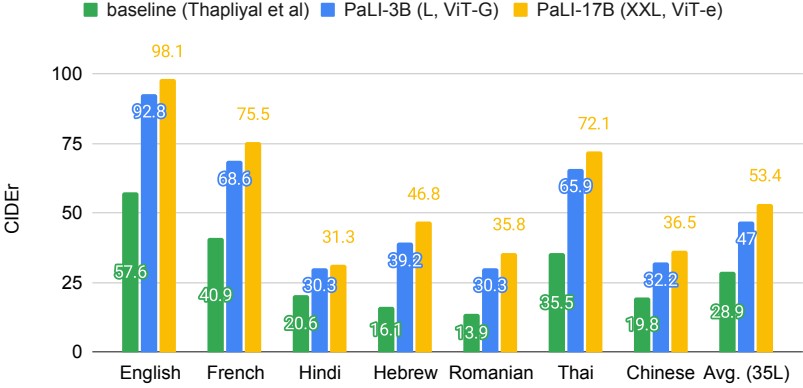

Translating non-English predictions back to English on 600 example subset

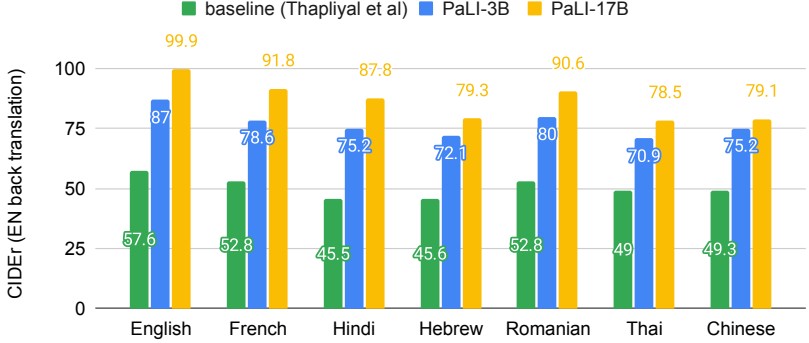

Figure 5: PaLI Scaling performance across multiple languages (See Table 2), using the Crossmodal-3600 benchmark. Larger scale models are important for better performance in these languages, especially low resource ones. (Top) CIDEr scores computed using predictions in each language. (Bottom) For the six languages French, Hindi, Hebrew, Romanian, Thai and Chinese, we sample a 600-example subset and back-translate the non-English predictions to English, and computed the CIDEr score vs. the same English references.

including French, Hindi, Hebrew, Romanian, Thai and Chinese to English and compute the CIDEr score against English references for a better comparison to the English quality. The result shows that the captioning quality across languages is fairly consistent.

We also trained a 5B PaLI model consisting of mT5-Large and ViT-e for additional datapoints. We evaluated this 5B model on two representative captioning and VQA benchmarks, COCO-Cap and OKVQA, and the results are shown in Table 12. We note that the training mixture and hyperparameters of this PaLI-5B checkpoint are slightly different from other PaLI sizes, but the results are still indicative and supportive of our conclusions regarding the value of joint scaling.

On COCO, the improvement from PaLI-3B to 5B (+2.1 CIDEr points) is slightly smaller than the improvement from PaLI-15B to 17B (+2.8). On OKVQA, it is likely that the benefit of having ViT-e cannot be exploited by the mT5-Large enc-dec as much as that by the mT5-XXL on VQA tasks, which require stronger language-understanding capabilities than Image Captioning tasks. In general, it is clear that scaling ViT still has much better return on investment (see the last column in Table 12), even for PaLI-5B where the ViT model is much larger than the encoder-decoder backbone. Note that we computed RoI as "improvement per 1B parameter", using COCO and OKVQA numbers as performance indicators.

| Model | Component | COCO-Cap @490 res | OKVQA @490 res | Improvement per 1B more params |
|-------|-----------|-------------------|----------------|--------------------------------|
| PaLI-3B | mT5-Large & ViT-G | 145.4 | 52.4 | - |
| PaLI-5B | mT5-Large & ViT-e | 147.5 | 53.8 | +0.9 per 1B more **ViT** params (vs PaLI-3B) |
| PaLI-15B | mT5-XXL & ViT-G | 146.2 | 56.5 | +0.2 per 1B more **mT5** params (vs PaLI-3B) |
| PaLI-17B | mT5-XXL & ViT-e | 149.0 | 62.4 | +2.2 per 1B more **ViT** params (vs PaLI-15B) +0.4 per 1B more **mT5** params (vs PaLI-5B) |

Table 12: Result on a 5B version of PaLI consisting of mT5-Large and ViT-e. Results on COCO-Cap and OKVQA with 490×490 are shown together with other sizes.

## C.3 ADDITIONAL ABLATIONS

Table 13 shows that initializing from unimodal checkpoint plays a critical role in PaLI's quality. Table 14 shows that freezing ViT during pretraining leads to an improvement in downstream finetuning on COCO.

Table 15 shows the effect of the non-English part of WebLI data. The table shows two sets of comparison for the pretraining data. 1) Using only the English subset of WebLI vs using only the whole WebLI. 2) Taking out the non-EN part of WebLI from the full mix vs using the full mix. This set of comparison results is performed with a 1.5B version of PaLI model, consisting of mT5-Large and ViT-L (with 300M parameters). This model has a similar parameter ratio (20% for ViT) compared with PaLI-17B (23%). Each model is pretrained to cover 200M of the data. All downstream benchmarks are fine-tuned and evaluated at 224×224 image resolution. The six non-En languages (6L) for XM-3600 are fr, hi, iw, ro, th and zh, and "7L" for xGQA are en, bn, de, id, ko, pt, ru, zh, both are the same as those included in Table 2 and Table 4. The takeaways are as follows:

- (comparison 1, row #1 vs row #2) With only the English portion of WebLI, the model's multilingual captioning capability remains very low (as measured on XM-3600), even with further finetuning on COCO-35L. There is also a clear drop in cross-lingual VQA performance on xGQA.

- (comparison 2, row #3 vs row #4) Taking away the multilingual part of WebLI from the full mixture, which still contains other translated multilingual/cross-lingual datasets (CC3M-35L, VQ2A-CC3M-35L, VQG-CC3M-35L), still has a significant impact on XM-3600 performance. On xGQA, because of the cross-lingual training source VQ2A-CC3M-35L, the impact of removing non-EN WebLI data is reduced but still apparent. With the non-EN WebLI data in the full mix, xGQA performance improves by +0.4 overall and is better than or equal to with only the WebLI-EN in every language.

- Last but definitely not least, there is an interesting result: when training with all the languages of WebLI, the model is performing better on (English) COCO captions, compared to training with English-only WebLI (about +2 CIDEr points). This suggests that 1) the multilingual WebLI may contain extra images with richer objects and their descriptions compared with the English-only subset 2) the model may be able to exploit the shared linguistic structure across languages, benefiting from transfer learning across languages.

| Model | Initialization | COCO (Karp. test) | XM-3600 | TextVQA |
|-------|----------------|-------------------|---------|---------|
| PaLI-3B | From mT5-Large and ViT-G | 141.4 | 93.8 (EN) / 42.5 (6L) | 41.6 |
| | From scratch | 72.8 | 22.1 (EN) / 10.1 (6L) | 12.8 |

Table 13: Comparison between PaLI's initializing from existing unimodal checkpoints and initializing the parameter from scratch. The setup is the same as the main ablation result Table 6.

| Model | ViT during finetuning | ViT during pretraining | COCO (Karp. test) |
|-------|----------------------|------------------------|-------------------|
| PaLI-3B | Fine-tuned | Frozen | 139.3 |
| | | Fine-tuned | 138.8 |
| PaLI-15B | Fine-tuned | Frozen | 141.4 |
| | | Fine-tuned | 140.1 |

Table 14: Comparison of performance on COCO for Frozen versus fine-tuned ViT during a short period of pretraining. In this comparison, finetuning of COCO is performed at resolution $224 \times 224$.

| Pretraining Data | XM-3600 (FT on COCO-35L) | COCO-Cap | xGQA (FT on VQAv2-13L) |
|------------------|--------------------------|----------|------------------------|
| only WebLI-en | 86.0 (en) / 8.2 (6L) | 132.2 | 40.6 (en) / 34.0 (7L) |
| only WebLI | 87.2 (en) / 30.0 (6L) | 134.3 | 42.8 (en) / 38.6 (7L) |
| WebLI-en & rest of PaLI mix | 91.2 (en) / 39.0 (6L) | 135.3 | 44.9 (en) / 40.9 (7L) |
| Full PaLI mix | 92.2 (en) / 41.9 (6L) | 135.4 | 45.1 (en) / 41.3 (7L) |

Table 15: Ablation studies on the effect of including the multilingual examples of WebLI on multi-(cross-)lingual benchmarks XM-3600 and xGQA. We also included the English benchmark COCO-Captions in the comparison. This set of comparison results is performed with a 1.5B version of PaLI model, consisting of mT5-Large and ViT-L (with 300M parameters).

## C.4 EVALUATION OF PALI'S VISUAL COMPONENT: VIT-e

Table 16 compares the ViT-e architecture with the smaller ViT-G and ViT-g architectures on vision only and vision-language tasks. The results suggest that V&L tasks could benefit more from scaling up the vision backbone, even on the high end. In Table 17, we fine-tune the pretrained ViT-e model on the ImageNet dataset, and then report the evaluation scores on several out-of-distribution test variants: ImageNet-v2, ObjectNet, and ReaL (Beyer et al., 2020). We follow the finetuning protocol of Zhai et al. (2022a), but use a $560 \times 560$ resolution. We evaluate the fine-tuned model at $644 \times 644$ (Touvron et al., 2019) (chosen according to a held-out 2% of the training set), results are reported in Table 17. ViT-e achieves 90.9% top-1 accuracy on ImageNet and shows clear benefits on the OOD benchmarks.

| | INet-10 | INet-25 | COCO | VQAv2 |
|---|---------|---------|------|-------|
| ViT-g | 84.5 | 85.4 | - | - |
| ViT-G | 84.9 | 85.6 | 146.2 | 82.9 |
| ViT-e | 85.2 | 85.8 | 149 | 83.4 |

Table 16: Impact of scaling ViT. For vision-only tasks, we report 10-shot and 25-shot accuracy on ImageNet. For vision-language tasks, ViT models are paired with the mT5-XXL model in PaLI and we report captioning (COCO) and VQA (VQAv2). For direct comparison, results with ViT-e on COCO and VQAv2 do not include the high resolution phase of pretraining.

Since ViT-e is new and has not been evaluated in the prior work, we evaluate its standalone performance. For this, we perform supervised fine-tuning on standard classification tasks. Additionally, we perform LiT transfer (Zhai et al., 2022b) to evaluate the frozen representation quality in a zero-shot setup.

We follow LiT (Zhai et al., 2022b) to add zero-shot transfer capabilities to the (frozen) ViT-e model, the visual component of PaLI. More specifically, we tune a text encoder, when the ViT image encoder is frozen. We use the English subset of the WebLI dataset for the text encoder training, since all evaluation tasks in Table 18 are in English. These results highlight that going from ViT-g to ViT-e

| Model | INet | INet-v2 | ObjNet | ReaL |
|-------|------|---------|--------|------|
| ViT-G | 90.5 | 83.3 | 70.5 | 90.8 |
| CoCa | 91.0 | - | - | - |
| ViT-e | 90.9 | 84.3 | 72.0 | 91.1 |

Table 17: ViT-e on ImageNet and OOD test sets.

| Model | INet | INet-v2 | INet-R | INet-A | ObjNet | ReaL | VTAB-N |
|---|---|---|---|---|---|---|---|
| CLIP (Radford et al., 2021) | 76.2 | 70.1 | 88.9 | 77.2 | 72.3 | - | 73.9 |
| ALIGN (Jia et al., 2021) | 76.4 | 70.1 | 92.2 | 75.8 | 72.2 | - | - |
| BASIC (Pham et al., 2021) | 85.7 | 80.6 | 95.7 | 85.6 | 78.9 | - | - |
| CoCa (Yu et al., 2022) | 86.3 | 80.7 | 96.5 | 90.2 | 82.7 | - | - |
| LiT ViT-g (Zhai et al., 2022b) | 85.2 | 79.8 | 94.9 | 81.8 | 82.5 | 88.6 | 74.7 |
| LiT ViT-e (ours) | 85.4 | 80.6 | 96.1 | 88.0 | 84.9 | 88.4 | 76.9 |

Table 18: Zero-shot transfer results of ViT-e on ImageNet, OOD test sets and VTAB-Natural datasets.

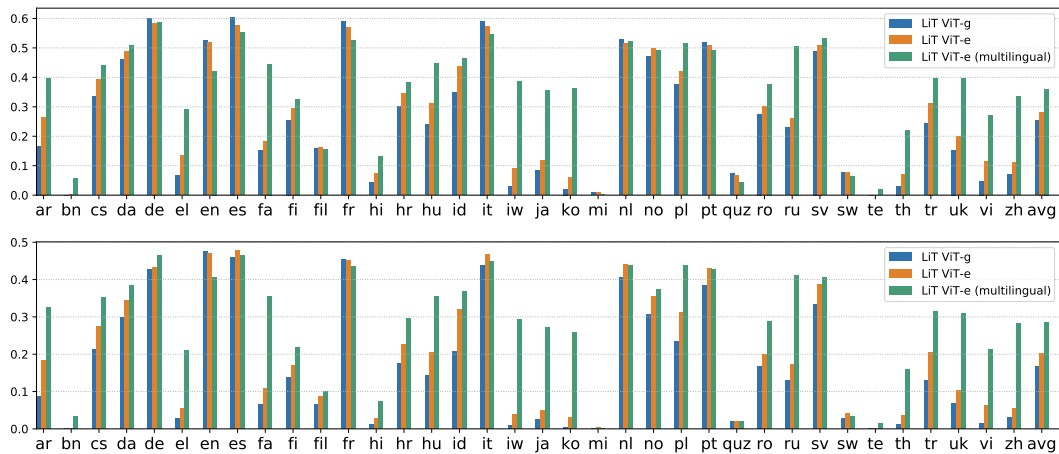

Figure 6: Zero-shot image-text retrieval results on all 36 languages of Crossmodal-3600. Top: image-to-text retrieval accuracy; bottom: text-to-image retrieval accuracy.

provides consistently better results. Notably, LiT with ViT-e achieves 84.9% zero-shot accuracy on the challenging out-of-distribution ObjectNet test set, setting the new state-of-the-art. The VTAB-Natural benchmark (Zhai et al., 2019) consists of seven diverse natural image datasets, for which LiT also benefits from ViT-e over ViT-g. Detailed results on each VTAB-Natural task are in Appendix C.6.

We also test multilingual performance using WebLI in this setting. We further perform LiT transfer using the same multilingual WebLI dataset as used to train PaLI, and use Crossmodal-3600 to evaluate the cross-lingual image-text retrieval performance. Figure 6 shows that LiT ViT-e pretrained on the English subset substantially outperforms the same model pretrained on the multilingual dataset. The same observation applies to a few languages that are similar to English, e.g. Spanish (es), French (fr), Italian (it). However, the multilingual model performs much better on most other languages, especially those with a non-latin script such as Chinese (zh), Japanese (ja), Korean (ko), and Hebrew (iw). On average (avg), the multilingual LiT ViT-e outperforms the English-only model by a large margin. More results could be found in Table 22. These results highlight the importance of having good multilingual benchmarks to measure the benefits of training models on diverse datasets such as WebLI.

## C.5 RESULTS ON TEXTCAPS, TEXTVQA AND VIZWIZ-QA WITHOUT DETECTED OCR AS INPUT

In the main text, we presented results on TextCaps, TextVQA, VizWiz-Cap, VizWiz-QA and ST-VQA with detected OCR strings as input. Following Kil et al. (2022), we order the OCR items based on their locations in the image, from top left to bottom right. We only include the OCR strings themselves, without the OCR-item locations provided by the API. GIT2 (Wang et al., 2022a) has demonstrated strong performance without the OCR input, while PaLI-17B shows the superiority of levaraging a specialized OCR system for a better recipe to solve these tasks.

Table 19 shows the results on TextCaps, TextVQA and VizWiz-QA without the detected OCR strings as input. PaLI slightly suffers without OCR input, while its performance remains close to the first version of GIT. This result may suggest that the significantly larger vocab of PaLI adds further difficulty to OCR string generation.

However, for VizWiz-QA, PaLI establishes SOTA performance without OCR input.

| Method | OCR input? | TextCaps test | TextVQA test | VizWiz-QA test-dev | VizWiz-QA test-std |
|---|---|---|---|---|---|
| TAP (Yang et al., 2021) | Yes | 103.2 | 53.97 | - | - |
| GIT | No | 138.2 | 59.75 | 68.0 | 67.5 |
| GIT2 | No | 145.0 | 67.27 | 71.0 | 70.1 |
| PaLI | No | 135.4 | 58.80 | 71.6 | 70.7 |
| PaLI | Yes | 160.4 | 73.06 | 74.4 | 73.3 |

Table 19: Results on TextCaps, TextVQA and VizWiz-QA with and without detected OCR as input for PaLI

## C.6 DETAILED VTAB RESULTS

For the VTAB benchmark (Zhai et al., 2019), we follow the methodology outlined in (Zhai et al., 2022b). PaLI sets a new state-of-the-art zero-shot performance for the "natural" subset (see Table 20).

| | Caltech101 | CIFAR-100 | DTD | Flowers102 | Pets | Sun397 | SVHN | Mean |
|---|---|---|---|---|---|---|---|---|
| CLIP | **92.8** | 77.5 | 55.7 | 78.3 | 93.5 | 68.4 | **51.0** | 73.9 |
| LiT *ViT-g* | 79.2 | 83.6 | 66.6 | **92.3** | 97.7 | 76.0 | 27.5 | 74.7 |
| LiT *ViT-e* | 79.8 | **90.4** | **68.8** | 91.2 | **98.1** | **76.3** | 33.8 | **76.9** |

Table 20: Accuracies for zero-shot evaluation of different VTAB "natural" tasks, and the average over these tasks. Note that CLIP is using OCR for the SVHN task (as opposed to LiT and PaLI, which do not use OCR).

## C.7 TOP 5 ACCURACY ON ZERO-SHOT IMAGENET DATASETS

Table 21 shows the Top 5 Accuracy results on Zero-shot evaluation on ImageNet Datasets.

| Model | INet | INet-R | INet-A | INet-sketch | INet-v2 | ObjNet |
|---|---|---|---|---|---|---|
| PaLI-3B | 84.31 | 90.05 | 55.04 | 76.47 | 78.49 | 53.71 |
| PaLI-15B | 84.78 | 90.91 | 59.00 | 76.81 | 79.54 | 55.29 |
| PaLI-17B | 86.18 | 91.51 | 62.72 | 79.30 | 80.71 | 58.35 |

Table 21: Top 5 accuracy results of Zero-shot image classification on ImageNet (Deng et al., 2009), ImageNet-R (Hendrycks et al., 2021a), ImageNet-A (Hendrycks et al., 2021b), ImageNet-Sketch (Wang et al., 2019b), ImageNet-v2 (Recht et al., 2019) and ObjectNet (Barbu et al., 2019).

## C.8 MORE ZERO-SHOT IMAGE-TEXT RETRIEVAL RESULTS ON CROSSMODAL-3600

Table 22 shows more zero-shot image-text retrieval results on Crossmodal-3600.

## D MODEL FAIRNESS, BIASES, AND OTHER POTENTIAL ISSUES

Models trained on web data are at risk of being biased or unfair due to biases in that data. A first step towards addressing those risks is being transparent about their existence, and then measuring them.

| Language | Image-to-text | | | Text-to-image | | |
|---|---|---|---|---|---|---|
| | LiT ViT-g | LiT ViT-e | LiT ViT-e (multilingual) | LiT ViT-g | LiT ViT-e | LiT ViT-e (multilingual) |
| ar | 5.28 | 26.58 | 39.69 | 2.80 | 18.46 | 32.60 |
| bn | 0.00 | 0.11 | 5.67 | 0.00 | 0.06 | 3.31 |
| cs | 18.19 | 39.25 | 44.03 | 11.24 | 27.35 | 35.24 |
| da | 26.44 | 48.92 | 50.75 | 14.07 | 34.43 | 38.48 |
| de | 37.83 | 58.42 | 58.53 | 23.61 | 43.25 | 46.50 |
| el | 1.56 | 13.47 | 29.03 | 0.39 | 5.46 | 20.92 |
| en | 51.22 | 51.78 | 42.11 | 46.24 | 47.07 | 40.63 |
| es | 41.81 | 57.50 | 55.22 | 30.29 | 47.71 | 46.55 |
| fa | 3.78 | 18.39 | 44.50 | 1.57 | 10.74 | 35.58 |
| fi | 14.14 | 29.42 | 32.64 | 6.59 | 16.91 | 21.80 |
| fil | 10.94 | 16.39 | 15.53 | 4.18 | 8.66 | 10.04 |
| fr | 38.28 | 57.06 | 52.61 | 28.02 | 45.20 | 43.47 |
| hi | 0.47 | 7.33 | 13.14 | 0.08 | 2.90 | 7.42 |
| hr | 15.86 | 34.47 | 38.31 | 8.80 | 22.72 | 29.55 |
| hu | 15.11 | 31.17 | 44.67 | 8.45 | 20.52 | 35.49 |
| id | 24.11 | 43.72 | 46.33 | 12.99 | 32.08 | 36.75 |
| it | 39.69 | 57.47 | 54.53 | 27.07 | 46.79 | 44.76 |
| iw | 1.75 | 9.11 | 38.67 | 0.86 | 3.99 | 29.39 |
| ja | 3.61 | 11.67 | 35.47 | 1.20 | 4.91 | 27.24 |
| ko | 1.78 | 6.00 | 36.11 | 0.35 | 3.14 | 25.95 |
| mi | 0.58 | 0.92 | 0.33 | 0.19 | 0.30 | 0.22 |
| nl | 37.47 | 51.67 | 52.14 | 27.26 | 44.08 | 43.79 |
| no | 26.53 | 49.69 | 49.17 | 14.61 | 35.59 | 37.35 |
| pl | 19.67 | 42.03 | 51.42 | 12.00 | 31.13 | 43.72 |
| pt | 33.92 | 50.81 | 49.19 | 23.58 | 42.97 | 42.73 |
| quz | 5.08 | 6.83 | 4.31 | 1.85 | 1.89 | 1.90 |
| ro | 17.94 | 30.08 | 37.75 | 10.15 | 20.06 | 28.82 |
| ru | 12.00 | 26.22 | 50.64 | 5.76 | 17.19 | 41.11 |
| sv | 25.50 | 51.00 | 53.22 | 15.11 | 38.80 | 40.66 |
| sw | 4.47 | 7.75 | 6.42 | 1.58 | 4.17 | 3.41 |
| te | 0.06 | 0.03 | 1.92 | 0.03 | 0.03 | 1.42 |
| th | 1.89 | 7.22 | 22.00 | 0.79 | 3.71 | 16.06 |
| tr | 10.72 | 31.28 | 39.50 | 4.73 | 20.42 | 31.47 |
| uk | 7.67 | 19.94 | 39.53 | 3.38 | 10.40 | 30.81 |
| vi | 3.08 | 11.44 | 27.08 | 0.98 | 6.22 | 21.28 |
| zh | 4.53 | 11.11 | 33.61 | 1.67 | 5.60 | 28.24 |
| avg | 15.64 | 28.23 | 35.99 | 9.79 | 20.14 | 28.46 |

Table 22: Image-to-text and text-to-image zero-shot retrieval results on all 36 languages of Crossmodal-3600. Models are trained following LiT (Zhai et al., 2022b) method with diverse visual backbones (ViT-g or ViT-e) and datasets (English or multilingual).

To this end, we add a data card (Pushkarna et al., 2022) for WebLI and model card (Mitchell et al., 2019) for PaLI in Appendix G and F.

To understand the demographic properties of the data, we sample 112,782 (0.001% of the full data set, randomly sampled due to the limitations of the labeling tool, described next) examples and analyze both images and texts of the sampled data with the Know Your Data (KYD) tool. We use KYD to analyze the perceived gender presentation of image subjects (Schumann et al., 2021) along with gender expressed through pronouns in text. In the sampled images, 54% of people appear feminine presenting with 46% masculine presenting. In the sampled text, female pronouns (e.g., she, her) are used 30% of the time, male pronouns (e.g., he, him) 38% of the time, and they or them (either singular or plural) 31% of the time. We also analyze the perceived age of individuals appearing in the sampled images, resulting in the distribution displayed in Figure 7.

We consider all the effort above a first step, and know that it will be important to continue to measure and mitigate bias as we apply our model to new tasks. Deeper analysis will include the study of the model's recognition capabilities and potential biases observed towards specific attributes, e.g. related to gender, age, etc. and how scaling affects these observations.

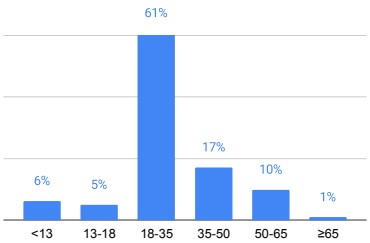

Figure 7: The distribution of ages recognized from the sampled images of WebLI.

# E  LIMITATIONS

Despite good performance, our model has a number of limitations. For example, the model might not describe very thoroughly a complex scene with many objects because most of the source data does not have complex annotations. We have tried to mitigate this with the object-aware and localization aware queries, added to the data.

We also noticed that some of the multilingual capabilities are lost when fine-tuned on English-only data, which is consistent with other model fine-tuning behavior. Ideally these models should be fine-tuned on a mix of multiple datasets including multilingual ones.

There are limitations related to the evaluation procedures of the benchmarks. Since we are evaluating in the open-vocabulary generative setting, for example in VQA, the model might generate a correct response which is a synonym or a paraphrase of the target response and does not match the target exactly. In these cases the answer is counted as incorrect. Fixed-vocabulary approaches do not suffer from these issues, but are limited in generalization beyond the answers of a specific dataset. Further, in terms of evaluation, some benchmarks might need more comprehensive strategies to avoid evaluations with Western-centric bias. Multilingual models and benchmarks are a first step in that direction.

# F  PaLI MODEL CARD

Following Mitchell et al. (2019), we present the PaLI model card in Table 23.

| Model Summary | |
|---|---|
| Model Architecture | PaLI is a multimodal sequence-to-sequence Transformer (Vaswani et al., 2017) model derived from the T5 (Raffel et al., 2020) encoder-decoder architecture. It takes text tokens and ViT (Dosovitskiy et al., 2021) dense image embeddings as inputs to an encoder and autoregressively predicts discrete text tokens with a decoder. |
| Input(s) | A pair of image and text. |
| Output(s) | Generated text. |
| **Usage** | |
| Application | The model is for research prototype and the current version is not available for the public. |
| Known Caveats | No. |
| **System Type** | |
| System Description | This is a standalone model. |
| Upstream Dependencies | No. |
| Downstream Dependencies | No. |
| **Implementation Frameworks** | |

| | |
|---|---|
| Hardware & Software | Hardware: TPU v4 (Jouppi et al., 2020).

Software: T5X (Roberts et al., 2022), JAX (Bradbury et al., 2018), Flaxformer (Heek et al., 2020)

Details are reported in Section A.1. |
| Compute Requirements | Reported in Section A.1. |
| **Model Characteristics** | |
| Model Initialization | The model is initialized from pre-trained language (mT5) (Xue et al., 2021) and Vision Transformer (ViT) (Zhai et al., 2022a; Dosovitskiy et al., 2021) checkpoints. |
| Model Status | This is a static model trained on an offline dataset. |
| Model Stats | The largest PaLI model has 17B parameters, which consists of a 13B parameter mT5-XXL model and a 4B parameter ViT-e model. We have also trained 3B and 15B parameter models. |
| **Data Overview** | |
| Training dataset | The model is pre-trained on the following mixture of datasets: WebLI (Table 24), CC3M-35L (Sharma et al., 2018), VQ$^2$A-CC3M-35L (Changpinyo et al., 2022a), Open Images (Kuznetsova et al., 2020), Visual Genome (Krishna et al., 2017) and Object365 (Shao et al., 2019). Details are reported in Section A.2. |

| Evaluation and Fine-tuning Dataset | |
|---|---|
| | • **Vision + language tasks** |
| |     – **Image captioning (English)**: COCO (Chen et al., 2015), NoCaps (Agrawal et al., 2019), TextCaps (Sidorov et al., 2020) |
| |     – **Image captioning (multilingual)**: Crossmodal-3600 (Thapliyal et al., 2022) |
| |     – **Visual question answering (English)**: VQAv2 (Goyal et al., 2017), OKVQA (Gui et al., 2021), TextVQA (Singh et al., 2019), VizWiz-QA (Gurari et al., 2018) |
| |     – **Visual question answering (multilingual)**: xGQA (Pfeiffer et al., 2022), MaXM (Changpinyo et al., 2022b) |
| | • **Vision-only tasks** |
| |     – **Image classification (fine-tuning)**: ImageNet (Deng et al., 2009), ImageNet-V2 (Recht et al., 2019), ObjectNet (Barbu et al., 2019), ReaL (Beyer et al., 2020) |
| |     – **Image classification (zero-shot)**: ImageNet (Deng et al., 2009), ImageNet-V2 (Recht et al., 2019), ImageNet-R (Hendrycks et al., 2021a), ImageNet-A (Hendrycks et al., 2021b), ImageNet-Sketch (Wang et al., 2019b), ObjectNet (Barbu et al., 2019), ReaL (Beyer et al., 2020), VTAB (Zhai et al., 2019) |
| | • **Language-only tasks** |
| |     – **Natural language inference (English)**: SuperGLUE (Wang et al., 2019a) |
| |     – Natural language inference (multilingual): XNLI (Conneau et al., 2018) |
| |     – **Question Answering (multilingual)**: XQuAD (Artetxe et al., 2020), TyDiQA (Clark et al., 2020) |

| **Evaluation Results** | |
|---|---|
| Evaluation Results | Reported in Section 4. |

| **Model Usage & Limitations** | |
|---|---|
| Sensitive Use | The model is capable of open-ended text generations. This model should not be used for any of the unacceptable language model use cases, e.g., generation of toxic speech. |
| Known Limitations | Reported in Section E. |
| Ethical Considerations & Risks | Reported in Section D. |

Table 23: PaLI model card.

## G  WEBLI DATASHEET

Following Gebru et al. (2021), we present the WebLI datasheet in Table 24.

| Motivation | |
|---|---|
| For what purpose was the dataset created? Who created the dataset? Who funded the creation of the dataset? | The dataset was created to support vision-language research, such as the large-scale pre-training for image understanding, image captioning, visual question answering, object detection etc. |
| Any other comments? | No user data is included in the data source. Personally identifiable and privileged data are filtered out during the dataset construction. |
| **Composition** | |
| What do the instances that comprise the dataset represent (e.g., documents, photos, people, countries)? | Each instance is presented as an image and associated texts (alt-text, page title and OCR) collected from the web. |
| How many instances are there in total (of each type, if appropriate)? | There are 9,624,017,440 instances in total (about 260 TB in size). |
| Does the dataset contain all possible instances or is it a sample (not necessarily random) of instances from a larger set? | The dataset is built from the public web pages. It is not a complete set but rather a subset of the publicly available image-text pairs. |
| What data does each instance consist of? | Each instance consists of 20+ features. Most features are from public web pages; a few are from publicly available automatic services. The primary features are image pixels and the associated texts, including alt-text, page title and OCR. Other features include rich image and page meta information (e.g. URL, MIME type) and filter signals (attached to alt-text only). |
| Is there a label or target associated with each instance? | No. |
| Is any information missing from individual instances? | No. |
| Are relationships between individual instances made explicit? | There are no relationships between individual instances. |
| Are there recommended data splits? | There is only one split containing all the instances of the dataset. |
| Are there any errors, sources of noise, or redundancies in the dataset? | The dataset is built from the web and only applied a few filters. The data is noisy and redundant images or texts may exist. |
| Is the dataset self-contained, or does it link to or otherwise rely on external resources? | The dataset is self-contained. |
| Does the dataset contain data that might be considered confidential? | No. |

| | |
|---|---|
| Does the dataset contain data that, if viewed directly, might be offensive, insulting, threatening, or might otherwise cause anxiety? | The dataset likely contains data that might be considered offensive, insulting or threatening as the data is collected from the web. We use algorithmic methods and classifiers to remove sensitive / personal identifiable information (PII) / pornographic images. |

### Collection Process

| | |
|---|---|
| How was the data associated with each instance acquired? | Images, alt-text and meta information are from the public web. Text language identification and OCR annotation are done via publicly available automatic services. |
| What mechanisms or procedures were used to collect the data? | The data was collected using a variety of pipelines, software programs and publicly available automatic services to extract and filter images and texts. |
| If the dataset is a sample from a larger set, what was the sampling strategy? | The dataset is built from a subset of public web pages. |
| Over what timeframe was the data collected? | 2021-2022 |
| Were any ethical review processes conducted? | No. |

### Preprocessing, cleaning, and labeling

| | |
|---|---|
| Was any preprocessing, cleaning, or labeling of the data done (e.g., discretization or bucketing, tokenization, part-of-speech tagging, SIFT feature extraction, removal of instances, processing of missing values)? | The dataset is not annotated. Images which are identified as having adult content are excluded. Empty texts and texts (alt-text, page title and OCR) which are identified as PII are excluded. Images identified as having adult content, with improper shape, or with too many paired-texts are excluded. |
| Is the software used to preprocess, clean, or label the instances available? | No. |

### Uses

| | |
|---|---|
| Has the dataset been used for any tasks already? | Yes, we use the dataset for pre-training PaLI models. |
| Is there a repository that links to any or all papers or systems that use the dataset? | No. |
| What (other) tasks could the dataset be used for? | Vision-only tasks (image classification, object detection etc.), language-only tasks (question answering, natural language inference etc.) and vision+Language tasks (image captioning, visual question answering, image-text retrieval etc.). |
| Is there anything about the composition of the dataset or the way it was collected and pre-processed/cleaned/labeled that might impact future uses? | The dataset is in a stable version and will be refreshed in the future to follow data policies. |
| Are there tasks for which the dataset should not be used? | The dataset should not be used for training any of the unacceptable vision, language or vision-language model use cases, e.g., generation of toxic captions or inappropriate images. |

### Distribution

| | |
|---|---|
| Will the dataset be distributed to third parties outside of the entity (e.g., company, institution, organization) on behalf of which the dataset was created? | No. |

Table 24: WebLI datasheet.

