# OpenReview forum: "PaLI: A Jointly-Scaled Multilingual Language-Image Model"
_ICLR.cc/2023/Conference — ICLR 2023 notable top 5%_

### Official Review · Reviewer_6grV · 2022-10-16

**Confidence:** 4
**Correctness:** 3
**Technical Novelty And Significance:** 3
**Empirical Novelty And Significance:** 2
**Recommendation:** 8

**Clarity, Quality, Novelty And Reproducibility:**

### Clarity

**The visual component, ViT-e.** It is unclear how it seamlessly changes the input resolution from 224x224 to others. Which part of representations of ViT-x is used for the input tokens to the Transformer encoder? Section 3.1 or Appendix A.1 should include this key detail for readers unfamiliar with this experimental setup.

**The name of PaLI.** This paper does not mention why they call it PaLI, which the readers only can assume -- It is for *Language and Image*, while reminiscent of the PaLM in the second paragraph of the Introduction.

### Novelty

The authors provide two major arguments on the joint scaling of the vision and language components and taking advantage of pre-training with large multilingual datasets as novel technical observations. However, for the reasons of W1 and W2 in the weakness section, their arguments are grounded on weak evidence to support their claims. Overall, the authors could be more carefully discussing the contributions of their model architecture and newly collected datasets by controlling other factors. The authors should resolve these issues in the rebuttal period for the recommendation of acceptance.

**N1. Resemblance to the Unified-IO.** The proposed VQA-like generalized task and the modular architecture with Transformer encoder/decoder are substantially similar to Unified-IO [1], but the authors fail to relate to, and properly compare with that. This work was initially published on June 17, 2022 (3-month before the ICLR deadline).

[1] Lu, J., Clark, C., Zellers, R., Mottaghi, R., & Kembhavi, A. (2022). Unified-IO: A Unified Model for Vision, Language, and Multi-Modal Tasks. http://arxiv.org/abs/2206.08916

### Reproducibility

Please see the weakness W3 about the exclusiveness in this study.

**Strength And Weaknesses:**

### Strength

This paper consistently shows state-of-the-art performance across multilingual image captioning, visual question answering, and zero-shot image classification tasks. Especially, the VQA performance in an open-vocabulary generation setting is inspiring.


### Weakness

**W1. Weak argument on the joint scaling of the vision and language components.** The third paragraph in Introduction argues that the visual component gives a better return on investment (RoI). Figure 2 seems to be the evidence for this claim; however, we cannot remove the possibility of other explanations since there is only one condition for each variable, the language side is from 1B to 13B (too wide interval) compared with that the visual side is from 2B to 4B. The return on investment can be non-linear. One suggestion is that we need the report of "PaLI-5B (L, ViT-e)" to see the RoI where the visual component cannot exploit the large language component. *Emphasize that, according to this result, one of two major claims can be rejected. The rating might be lowered if there is no reasonable explanation or experimental support.*

**W2. Weak support on multilingual pre-training.** Since the proposed models are pre-trained with multilingual datasets, we expect them reasonably works on multilingual downstream tasks. Table 2 and Table 4 Right (MaXM) only show the multilingual downstream performances, while Table 4 Left shows the cross-lingual for English-answers only (xGQA). However, this xGQA experimental setup does not firmly support the need for multilingual pre-training in cross-lingual tasks since the model architecture is not controlled (MPT may underperform because of model architecture or pre-trained datasets; we cannot discern by this experiment). Here, one suggestion is that, if PaLI-17B is pre-trained with the subset of the English-only WebLI, this model will significantly underperform the (original) multilingual PaLI-17B for the cross-lingual xGQA benchmark (even exploiting translations for multilingual tasks)?

**W3. Exclusiveness in the study.** The authors do not intend to release the dataset and pre-trained models considering the risk of exposure to "unknown biases or stereotypes, or propagate inaccurate or otherwise distorted information." In this vein, how can the community reproduce the results? The current *Reproducibility Statements* do not discuss this issue and the impact on research communities studying this topic, seriously enough.


**Summary Of The Paper:**

This is a technical paper exploring a couple of new directions toward a state-of-the-art *enormous*-scale multilingual vision-and-language model. There are two novel and interesting observations: 1) joint scaling of the vision and language components and 2) taking advantage of pre-training with large multilingual datasets for multilingual downstream tasks.

**Summary Of The Review:**

The proposed method shows strong performances across multilingual multimodal and unimodal downstream tasks. As a technical paper with empirical explorations, the authors tried to show two major observations; however, the two claims are weakly supported by experiments. In the weakness section, the suggestions to resolve these issues can be considered to improve the quality of the manuscript.

---

After reading the author's feedback, the raised issues seem to be sufficiently resolved.
I decide to raise the score recommending to accept.

---

> ### Author Response · Authors · 2022-11-18
> **Response to Reviewer 6grV (1/2)**
>
> **[On the joint scaling of the vision and language components]**
>
> We thank the reviewer for their insightful comments and suggestions. We did produce a PaLI-5B model earlier during the exploration phase of our project. Given the limited time and resource, we present here the results using this PaLI-5B checkpoint on two representative tasks: Image Captioning (COCO) and VQA (OK-VQA). We note that the training mixture and hyperparameters of this PaLI-5B checkpoint are slightly different from other PaLI sizes, but the results are still indicative and supportive of our conclusions regarding the value of joint scaling.
>
> | Model | Component | COCO-Cap @490 res | OKVQA @490 res | RoI (improvement per 1B more parameter)|
> |-|-|-|-|-|
> |PaLI-3B| mT5-Large & ViT-G/14 | 145.4 | 52.4 | - |
> |PaLI-5B| mT5-Large & ViT-e/14 | 147.5 | 53.8 | +0.9 per 1B increased **ViT** params (vs PaLI-3B) |
> |PaLI-15B| mT5-XXL & ViT-G/14 | 146.2 | 56.5 | +0.2 per 1B increased **mT5** params (vs PaLI-3B) |
> |PaLI-17B| mT5-XXL & ViT-e/14 | 149.0 | 62.4 | +2.2 per 1B increased **ViT** params (vs PaLI-15B) & +0.4 per 1B increased **mT5** params (vs PaLI-5B) |
>
> On COCO, the improvement from PaLI-3B to 5B (+2.1 CIDEr points) is slightly smaller than the improvement from PaLI-15B to 17B (+2.8). On OK-VQA, as the reviewer pointed out, it is likely that the benefit of having ViT-e cannot be exploited by the mT5-Large enc-dec as much as that by the mT5-XXL on VQA tasks, which require stronger language-understanding capabilities than Image Captioning tasks. In general, it is clear that scaling ViT still has much better Return on Investment (see the last column in the table above), even for PaLI-5B where the ViT model is much larger than the encoder-decoder backbone. Note that we computed RoI as “improvement per 1B parameter”, using COCO and OKVQA numbers as performance indicators.
>
> **[On multilingual pre-training]**
>
> First of all, we thank the reviewer for suggesting this ablation on multilingual training, as it proved very insightful and we will add these new findings to the paper. We produced a comprehensive ablation on the effect of multilingual pretraining by including other reviewer's suggestion.
>
> | Pretraining data | XM-3600 CIDEr (Fine-tune on COCO-35L) | COCO-Karp. CIDEr | xGQA accuracy (Fine-tune on VQAv2-13L) |
> | - | - | - | - |
> |WebLI-EN | 86.0 (en) / 8.2 (6L) | 132.2 | 40.6 (en) / 34.0 (7L) |
> |WebLI | 87.2 (en) / 30.0 (6L) | 134.3 | 42.8 (en) / 38.6 (7L) |
> |Full mix, w/ WebLI-EN | 91.2 (en) / 39.0 (6L) | 135.3 | 44.9 (en) / 40.9 (7L) |
> |Full mix | 92.2 (en) / 41.9 (6L) | 135.4 | 45.1 (en) / 41.3 (7L) |
>
> The table above shows two sets of comparison.
> 1. Using the English subset of WebLI vs using the whole  WebLI.
> 2. Taking out the non-EN part of WebLI from the full mix vs using the full mix.
>
> Given the limited time and resources, we trained these models with a new PaLI-1.5B model, consisting of mT5-large (1.2B) and ViT-L/16 (300M). This model has a similar parameter ratio (20% for ViT) compared with PaLI-17B (23%). Each model is trained to cover 200M of the data. All downstream benchmarks are fine-tuned and evaluated at 224x224 image resolution. The six non-En languages (6L) for XM-3600 are fr, hi, iw, ro, th and zh, and "7L" for xGQA are en, bn, de, id, ko, pt, ru, zh, both are the same as those included in Table 3 and Table 4 of the paper. The takeaways are as follows:
> - (comparison 1, row #1 vs row#2) With only the English portion of WebLI, the model’s multilingual captioning capability remains very low (as measured on XM-3600), even with further finetuning on COCO-35L. There is also a clear drop in cross-lingual VQA performance on xGQA.
> - (comparison 2, row #3 vs row#4) Taking away the multilingual part of WebLI from the full mixture, which still contains other translated multilingual/cross-lingual datasets (CC3M-35L, VQ2A-CC3M-35L, VQG-CC3M-35L), still has a significant impact on XM-3600 performance. On xGQA, because of the cross-lingual training source VQ2A-CC3M-35L, the impact of removing non-EN WebLI data is reduced but still apparent. With the non-EN WebLI data in the full mix, xGQA performance improves by +0.4 overall and is better than or equal to with only the WebLI-EN in every language.
> - Last but definitely not least, there is an interesting result: when training with all the languages of WebLI, the model is performing better on (English) COCO captions, compared to training with English-only WebLI (about +2 CIDEr points). This suggests that 1) the multilingual WebLI may contain extra images with richer objects and their descriptions compared with the English-only subset 2) the model may be able to exploit the shared linguistic structure across languages, benefiting from  transfer learning across languages.

---

> > ### Author Response · Authors · 2022-11-18
> > **Response to Reviewer 6grV (2/2)**
> >
> > **[On exclusiveness in the study]**
> >
> > We tried our best to provide detailed descriptions on construction of the dataset, the training mixture, and how the models are trained. Releasing the dataset and/or models at this scale is a challenging problem for the whole community; similar works such as SimVLM, Flamingo and GIT face the same issue. Besides the enormous storage and distribution cost, the potential misuse and the amplification of unwanted biases made by making the data public are both major concerns, and more work needs to be done. On the model side, the mT5 checkpoints we used are publicly available and we are actively working on sharing some of the remaining artifacts in the near future.
> >
> > **[On ViT-e and resolution]**
> >
> > Regarding the question of “Which part of representations of ViT-x is used for the input tokens to the Transformer encoder?”, this is in section 3.1 that we use the “output patch features of a Vision Transformer”. We will make this more clear.
> >
> > When scaling up the image resolution, the patch size is kept the same, and, as a result, the number of patches increases. We perform a 2D bilinear upsampling of the positional embedding to match the increased number of patches [cite the vit paper]. We added this detail into to Appendix A.
> >
> > **[On Unified-IO]**
> >
> > We thank the reviewer for pointing us to the concurrent work. We are now citing and discussing the Unified-IO paper in the Related Work and Architecture sections.

---

### Official Review · Reviewer_h6TT · 2022-10-24

**Confidence:** 4
**Correctness:** 4
**Technical Novelty And Significance:** 3
**Empirical Novelty And Significance:** 3
**Recommendation:** 8

**Clarity, Quality, Novelty And Reproducibility:**

The above work is very clear and of high quality demonstrating importance of scaling vision and language components for significant improvement in performance on existing benchmarks. However there is limited novelty as the work shows results of scaling on a new dataset

In terms of reproducibility it is unclear how to reproduce the WebLI dataset which is important for pretraining PALI model. Similarly, the best performing model, VIT-e is trained on JFT-3B dataset which is also difficult to reproduce. Authors also has not mentioned about releasing code for this work.

**Strength And Weaknesses:**

### Strengths
- The paper proposes an effective scaling strategy that significantly improves downstream performance on wide array of multimodal(vision+language) tasks
- The VIT-e model trained for this work seems to be really helping multimodal tasks and will be a very useful contribution to the community if released
- Experiments are quite extensive and thorough. Paper is well written and detailed.
- The paper does a thorough ablation on the different pre-training mixtures and usefulness of the different objectives on downstream tasks, which is valuable.


### Questions
- I am slightly concerned with the technical novelty of the work as it is widely known that most of the multimodal benchmarks continue to perform well when the vision backbone/component of the model is better.
- Language only tasks seem to have lower performance compared to T5XXL base model even with PaLI-17B. How does this change with smaller PaLI models where there is a smaller vision model?
- It is unclear what is the impact of multilingual data and would have been great to see ablation with and without using multilingual data.
- The performance on some of the datasets like VQA are way above human performance of ~81, which is a bit surprising. On tasks like VQA it has been evident from some time that better visual models have room for improvement, however to improve further, larger models must also get ambiguous answers correct. Although we are deduplicating and removing any overlap between pretraining and text data, do you think it might be possible there is some leakage, not in a full sample form, but either only on text side or image side? Can you provide more details about the deduplication?

**Summary Of The Paper:**

This paper proposes and effective strategy to jointly scale vision and language models that works on vision, language and vision & language multimodal tasks. The model is trained on a mixture of several pretraining tasks on a new dataset containing 1.6B samples in over 100 different languages. The model is simple and modular and achieves good performance on several multimodal tasks.

**Summary Of The Review:**

This work clearly shows the benefit of scaling both language and vision specific parameters in vision-language models. The work supports the findings through detailed experimentation and outperforms existing state-of-the-art methods by a significant margin. There is however not significant novelty in the methods but the findings are important for the community and hence my current rating. Looking forward to the author discussion phase.

---

> ### Author Response · Authors · 2022-11-18
> **Response to Reviewer h6TT (1/3)**
>
> **[novelty on better/larger vision backbone helps]**
>
> We agree with the reviewer that there have been works demonstrating improved performance when the vision backbone gets larger and better, for example, GIT2 [1]. However, compared with GIT, where the majority of the model parameters are in the vision encoder, we demonstrate the effectiveness of jointly scaling both vision and language components and also show that scaling up the vision component is more effective across a large parameter regime.
>
> Also, in our response to Reviewer 6grV, we show the results of a new PaLI-5B – consisting of a 1.2B mT5-large component and the 4B ViT-e component. ViT-e shows less advantage over ViT-G with this smaller encoder-decoder backbone. This adds to the support of our argument that the joint scaling of both vision and language backbones is important to multimodal performance, while scaling the vision backbone is more effective.
>
> [1] https://arxiv.org/abs/2205.14100
>
> **[Language only performance with a smaller PaLI]**
>
> | Model | XNLI | XQuAD |
> | - | - | - |
> | | ZS (Accu.) | ZS (F1/EM) |
> | mT5-Large (in paper [2]) | 81.1 | 77.8/61.5 |
> | mT5-Large (our repeat) | 80.7 | 77.9/61.5 |
> | PaLI-3B | 79.2 | 75.7/58.4 |
>
> The reviewer correctly pointed out that the language performance from PaLI-17B is slightly lower than the mT5-XXL backbone for some tasks, but overall the performance remains quite close to mT5-XXL. We want to note that in the PaLI training mixture, language-only span corruption only consists of 1/16 of the examples -- the model is focusing much more on Multimodal tasks than on text-only tasks. We do not have resources available to ablate much on the mixing ratio, but we believe that the gap can be closed by increasing the weight of language-only training examples in the mixture.
>
> Regarding the reviewer’s question, PaLI-15B and PaLI-17B use the same mT5-XXL backbone. Although PaLI-15B has a smaller ViT model, the ViT backbone in both cases is frozen during training, and the trainable parameters in both cases are from the same mT5-XXL backbone. Thus, we do not expect a systematic difference of the language-only capability between PaLI-15B and PaLI-17B.
> Given the resource available to us, we evaluated the PaLI-3B (with mT5-Large backbone) on XNLI and XQuAD, using the same zero-shot transfer setting as in the paper. We are able to  reproduce the performance of mT5-Large presented in the mT5 paper. However, unlike PaLI-17B which gets +0.4 higher than our reproduced numbers for mT5-XXL on XNLI and -0.7 lower on XQuAD, PaLI-5B is -1.5 lower on XNLI and -2.6 lower on XQuAD. This behavior is expected, due to the much smaller model capability of mT5-Large and much higher difficulty for it to maintain language-only understanding capability while gaining multimodal capabilities.
>
> [2] https://arxiv.org/abs/2010.11934

---

> > ### Author Response · Authors · 2022-11-18
> > **Response to Reviewer h6TT (2/3)**
> >
> > **[Ablation without using multilingual data]**
> >
> > | Pretraining data | XM-3600 CIDEr (Fine-tune on COCO-35L) | COCO-Karp. CIDEr | xGQA accuracy (Fine-tune on VQAv2-13L) |
> > | - | - | - | - |
> > |WebLI-EN | 86.0 (en) / 8.2 (6L) | 132.2 | 40.6 (en) / 34.0 (7L) |
> > |WebLI | 87.2 (en) / 30.0 (6L) | 134.3 | 42.8 (en) / 38.6 (7L) |
> > |Full mix, w/ WebLI-EN | 91.2 (en) / 39.0 (6L) | 135.3 | 44.9 (en) / 40.9 (7L) |
> > |Full mix | 92.2 (en) / 41.9 (6L) | 135.4 | 45.1 (en) / 41.3 (7L) |
> >
> > First of all, we thank the reviewer for suggesting this ablation on multilingual training, as we will add these new findings to the paper.
> > The table above shows two sets of comparison.
> > 1. Using the English subset of WebLI vs using the whole  WebLI.
> > 2. Taking out the non-EN part of WebLI from the full mix vs using the full mix.
> >
> > Given the limited time and resources, we trained these models with a new PaLI-1.5B model, consisting of mT5-large (1.2B) and ViT-L/16 (300M). This model has a similar parameter ratio (20% for ViT) compared with PaLI-17B (23%). Each model is trained to cover 200M of the data. All downstream benchmarks are fine-tuned and evaluated at 224x224 image resolution. The six non-En languages (6L) for XM-3600 are fr, hi, iw, ro, th and zh, and "7L" for xGQA are en, bn, de, id, ko, pt, ru, zh, both are the same as those included in Table 3 and Table 4 of the paper. The takeaways are as follows:
> > - (comparison 1, row #1 vs row#2) With only the English portion of WebLI, the model’s multilingual captioning capability remains very low (as measured on XM-3600), even with further finetuning on COCO-35L. There is also a clear drop in cross-lingual VQA performance on xGQA.
> > - (comparison 2, row #3 vs row#4) Taking away the multilingual part of WebLI from the full mixture, which still contains other translated multilingual/cross-lingual datasets (CC3M-35L, VQ2A-CC3M-35L, VQG-CC3M-35L), still has a significant impact on XM-3600 performance. On xGQA, because of the cross-lingual training source VQ2A-CC3M-35L, the impact of removing non-EN WebLI data is reduced but still apparent. With the non-EN WebLI data in the full mix, xGQA performance improves by +0.4 overall and is better than or equal to with only the WebLI-EN in every language.
> > - Last but definitely not least, there is an interesting result: when training with all the languages of WebLI, the model is performing better on (English) COCO captions, compared to training with English-only WebLI (about +2 CIDEr points). This suggests that 1) the multilingual WebLI may contain extra images with richer objects and their descriptions compared with the English-only subset 2) the model may be able to exploit the shared linguistic structure across languages, benefiting from  transfer learning across languages.
> >
> > **[Exceeding human performance in VQA]**
> >
> > We also believe (similar to the point made by the reviewer) that some of these benchmarks are probably close to saturation; however, the human performance is not necessarily always the ultimate ceiling. More specifically to VQA, the computation of human performance there is not perfect, as it uses other annotators' answers as reference (which are not guaranteed to be exhaustive, and therefore a perfectly good answer could still be considered “wrong”).
> > We further note that recent models (cited in the paper, e.g., BeIT) perform very well and already outperform the human performance, at Acc 84 (slightly underperforming ours). One main advantage of our approach is that it achieves these levels of performance with an open vocabulary setting, rather than picking from a set of predefined responses.
> >
> > Moreover, we see models exceeding the “human performance” by a large margin on Image Captioning tasks:
> > Both GIT and us exceed human performance by a large margin on TextCaps (CIDEr scores are: GIT2: 145; [2] PaLI-17B: 160; Human: 125 [2])
> > Multiple existing models outperform the human CIDEr for COCO-Captions (80-ish) and for  nocaps(85.3-ish)
> > Again, we consider this to be an artifact of how “human performance” is computed, which makes such ceilings informative but not necessarily unbreakable.
> >
> > For deduplication, we followed previous work (ALIGN [3]) to perform near-deduplication. A detailed description of the deduplication algorithm was provided in the referenced paper. We have added this citation to the paper as well.
> >
> > [2] https://arxiv.org/pdf/2205.14100.pdf
> > [3] https://arxiv.org/pdf/2102.05918.pdf

---

> > > ### Author Response · Authors · 2022-11-18
> > > **Response to Reviewer h6TT (3/3)**
> > >
> > > **[General applicability of scaling property obtained on WebLI-based data mixture]**
> > >
> > > We believe, as in similar works, the scaling behavior is generally applicable.
> > >
> > > **[Reproducibility]**
> > >
> > > We tried our best to provide detailed descriptions on construction of the dataset, the training mixture, and how the models are trained. For the WebLI raw data collection, we followed a process to those reported in previous works [1]. In Appendix B, and also the data card, we provide abundant details on this. We believe the information can be very helpful to experts who would like to create a dataset using a similar method at a similar scale.
> > >
> > > Releasing the dataset and/or models at this scale is a challenging problem for the whole community; similar works such as SimVLM, Flamingo and GIT face the same issue. Besides the enormous storage and distribution cost, the potential misuse and the amplification of unwanted biases made by making the data public are both major concerns, and more work needs to be done. On the model side, the mT5 checkpoints we used are publicly available and we are actively working on sharing some of the remaining artifacts in the near future.
> > >
> > > [1] https://arxiv.org/abs/2102.05918

---

### Official Review · Reviewer_4Maa · 2022-10-25

**Confidence:** 4
**Correctness:** 4
**Technical Novelty And Significance:** 3
**Empirical Novelty And Significance:** 4
**Recommendation:** 8

**Clarity, Quality, Novelty And Reproducibility:**

The paper is well-written and the proposed ideas are clearly described.  Its narrative was easy to follow and the main claims were supported by extensive empirical results.

In terms of model novelty, the technical contribution is rather limited as it re-uses components that are already available. The novel contributions of this work include the scaling of language and vision components, reaching new state-of-the-art in several tasks, and coming up with an effective multilingual pre-training task for training such models.

**Strength And Weaknesses:**

**Strengths**

The proposed encoder-decoder model design for language and vision tasks is simple and leverages existing pre-trained models to reduce training costs.

Finds that scaling the vision component leads to better accuracy per parameter/FLOP compared to language components.

PaLI outperforms the state-of-the-art on a large array of language and vision tasks such as image captioning, visual QA, language understanding, and zero-shot image classification. Notably, it even outperforms fixed-vocabulary approaches using an open-vocabulary generative setting.

**Weaknesses**

Analysis of the computational cost & time required for training and comparison with other models such as Flamingo are missing.

The paper shares model and dataset cards, however, it is not clear if the actual artifacts will be shared publicly which can hinder reproducibility and lead to wasted time for other researchers.

Other:
-  When comparing to other models, it would be useful for the reader to show the size of the baselines. E.g. Tables 1 and 2.

- What are the 1-shot and 5-shot performances of PaLi on the INet dataset? It reaches a very close performance to Flamingo even with a 0-shot but it would be interesting to also show if it can outperform the 5-shot version with a few examples.


**Summary Of The Paper:**

This paper proposes a new, joint encoder-decoder model for language and vision that leverages existing pre-trained encoder-decoder language models and vision transformers. The authors train the model on a large multilingual dataset containing 10B images and texts from 100 languages and investigate the joint scaling of the pre-trained components. Experiments show state-of-the-art results on several language and vision tasks, and the importance of scaling the vision component beyond the previous-largest.

**Summary Of The Review:**

Overall, this paper proposes a joint model for language and vision based on pre-trained components that reaches state-of-the-art in several downstream tasks. Key to its success is appropriate scaling and multilingual pre-training on a large number of tasks. Even though the artifacts are not shared, it provides models and data cards that can help interested researchers to reproduce the results.

---

> ### Author Response · Authors · 2022-11-18
> **Response to Reviewer 4Maa**
>
> **[computational cost & time]**
>
> We thank the reviewer for the suggestion. We provide the estimated FLOPS of PaLI models compared to other competitive models, e.g., Flamingo and GIT2 below. “*” indicates the values estimated by us given the description in the corresponding papers. The large FLOPS we estimated for GIT2 is due to the fact that it is trained on a huge amount of data (12.9B image-text pairs). We will add this table to the paper.
>
> | Model | Pretraining Data | Pretraining compute (PFLOPS*days) |  COCO & VQAv2 finetuning (PFLOPS*days) | TextVQA finetuning (PFLOPS*days) |
> | - | - | - | - | - |
> | PaLI-3B | 1.6B | 56 | 1.1 | 0.2 |
> | PaLI-15B | 1.6B | 434 | 3.7 | 0.8 |
> | PaLI-17B | 1.6B | 453 | 4.5 | 0.9 |
> | Flamingo (80B) | 2.3B | 1381 * | N/A | N/A |
> | GIT2 (5.1B) | 12.9B | 5513 * | N/A | N/A |
>
> **FLOPS estimation details**  We follow the approximation adopted by the GPT-3 [1] paper. Each token on each model parameter counts as 2 FLOPS (multiplication and addition) for the forward pass (ignoring attention). The backward pass is twice as expensive.
>
> [1] https://arxiv.org/pdf/2005.14165.pdf
>
>
> **[sharing artifacts]**
> We tried our best to provide detailed descriptions on construction of the dataset, the training mixture, and how the models are trained. Releasing the dataset and/or models at this scale is a challenging problem for the whole community; similar works such as SimVLM, Flamingo and GIT face the same issue. Besides the enormous storage and distribution cost, the potential misuse and the amplification of unwanted biases made by making the data public are both major concerns, and more work needs to be done. On the model side, the mT5 checkpoints we used are publicly available and we are actively working on sharing some of the remaining artifacts in the near future.
>
>
> **[Other improvements to the presentation of results]**
>
> [model comparison] Thanks for the suggestion. We have added size information of the other models in Tables where applicable.
>
> [1-shot and 5-shot] Thanks for the suggestion. We plan to study the few-shot and prompting capability systematically in follow-up work. Few-shot and prompting capabilities are not our main focus of the current paper. Designing and optimizing a model's few-shot and prompting capabilities is a non-trivial effort, which we think is more suitable to be adequately addressed in a future paper.

---

### Official Review · Reviewer_ErFD · 2022-11-02

**Confidence:** 4
**Correctness:** 4
**Technical Novelty And Significance:** 2
**Empirical Novelty And Significance:** 3
**Recommendation:** 6

**Clarity, Quality, Novelty And Reproducibility:**

This paper is written in high-quality with clear demonstrations and discussions. The work is hard to be reproduced because of its high computational cost and massive data requirements. Nevertheless, this work provides detailed training and testing settings.

**Strength And Weaknesses:**

Strengths:
- This work demonstrates a new multilingual pretrained vision-language model with the following distinguished properties: SOTA performance on multimodal tasks without catastrophic forgetting language-only understanding capabilities; multilingual pretraining while maintaining SOTA performance on English-only tasks.
- An interesting empirical insight that may benefit future VL pretraining: large-scale vision pretraining may not benefit vision-only tasks, but could improve VL pretraining tasks by a large margin.
- Sufficient in-depth analysis and ablation studies are performed for better model understanding.

Weaknesses:
1. Similar architecture for vision-language pretraining has early been proposed in various works, for example in [a]. To the best of my knowledge, the major difference is to use encoder-decoder architecture (mT5) for downstream VL pretraining.

[a] MERLOT: Multimodal Neural Script Knowledge Models. NeurIPS 2021.

2. Intuitively, the pre-trained model can be used for multimodal machine translation task. However, most tasks are still traditional VL tasks, such as VQA.

3. Both image captioning Sec. 4.1 and VQA Sec 4.2 require extra finetuning on downstream datasets. What are the results of zero-shot performance? Is it possible to generate target answers with few-shot demonstrations by prompting?

4. Can you show training and inference time duration comparisons with current VL pre-trained models across different scales of the proposed model?

**Summary Of The Paper:**

This work introduces PaLI, a new large-scale vision-language pretraining model with multilingual enhancement. The architecture follows the previous scheme that leverages language pretraining as the main component which takes in vision and language feature tokens. The main contributions of this work are three folds: A new large-scale dataset on multilingual image-language pretraining, in-depth analysis of multilingual vision-language pretraining, pretrained models of different scales with SOTA performance on downstream tasks.

**Summary Of The Review:**

This work provides a new multilingual multimodal pre-trained language model. The learning framework is not entirely new, yet some empirical insights are provided with sufficient experiments. This work may benefit future VL Pretraining or multilingual pretraining.

---

> ### Author Response · Authors · 2022-11-18
> **Response to Reviewer ErFD**
>
> **[Adding discussion on similar architecture for vision-language pretraining]**
>
> We thank the reviewer for pointing out MERLOT. Indeed, there are many previous works with a similar architecture. We have added MERLOT to the discussion of related work.
>
> **[multimodal machine translation task]**
>
> Thank you for this suggestion, and we also find exploring multilingual machine translation very interesting. In this work, our scope is to first establish solid results on the multilingual version of the more traditional V&L tasks, such as captioning and VQA. We believe PaLI is well-positioned to explore this task in the near future. Given the limited amount of time and resources available to us in the response period, we are unable to get this task done with the same quality as the rest of the results in the paper. If we’ll be able to generate such results before the deadline for the final version, we will be happy to report such results.
>
> **[few-shot and prompting]**
>
> We thank the reviewer for the suggestion. To answer the reviewer’s question, it is possible to generate target answers with few-shot demonstrations by prompting. However, similar to the above point, few-shot and prompting capabilities are not our main focus of the current paper. Designing and optimizing a model's few-shot and prompting capabilities is a non-trivial effort, which we think is more suitable to be adequately addressed in a future paper.
>
> **[Training and inference time comparison]**
>
> We provide the estimated FLOPS of PaLI models compared to other competitive models, e.g., Flamingo and GIT2 below. “*” indicates the values estimated by us given the description in the corresponding papers. The large FLOPS we estimated for GIT2 is due to the fact that it is trained on a huge amount of data (12.9B image-text pairs). We will add this table to the paper.
>
> | Model | Pretraining Data | Pretraining compute (PFLOPS*days) |  COCO & VQAv2 finetuning (PFLOPS*days) | TextVQA finetuning (PFLOPS*days) |
> | - | - | - | - | - |
> | PaLI-3B | 1.6B | 56 | 1.1 | 0.2 |
> | PaLI-15B | 1.6B | 434 | 3.7 | 0.8 |
> | PaLI-17B | 1.6B | 453 | 4.5 | 0.9 |
> | Flamingo (80B) | 2.3B | 1381 * | N/A | N/A |
> | GIT2 (5.1B) | 12.9B | 5513 * | N/A | N/A |
>
> **FLOPS estimation details**  We follow the approximation adopted by the GPT-3 [1] paper. Each token on each model parameter counts as 2 FLOPS (multiplication and addition) for the forward pass (ignoring attention). The backward pass is twice as expensive.
>
> [1] https://arxiv.org/pdf/2005.14165.pdf
>
> **[releasing dataset and model for reproducibility]**
>
> We tried our best to provide detailed descriptions on construction of the dataset, the training mixture, and how the models are trained. Releasing the dataset and/or models at this scale is a challenging problem for the whole community; similar works such as SimVLM, Flamingo and GIT face the same issue. Besides the enormous storage and distribution cost, the potential misuse and the amplification of unwanted biases made by making the data public are both major concerns, and more work needs to be done. On the model side, the mT5 checkpoints we used are publicly available and we are actively working on sharing some of the remaining artifacts in the near future.

---

### Public Comment · ~Dinh_Anh_Vu1 · 2023-02-14
**Why is `<extra_id_0>` added at the end of prompt?**

In the appendix, section A.2 The pretraining task mixture, I notice that prompts for almost tasks have `<extra_id_0>` at the end. For example, in the split-captioning, the prompt is "Generate the alt_text in `<lang>` at `<pos>`:  `<cap1>` `<extra_id_0>`". What is/are the purpose(s) of `<extra_id_0>` ?

---

### Decision · Program_Chairs · 2023-01-20

**Decision:**

Accept: notable-top-5%

**Justification For Why Not Higher Score:**

N/A

**Justification For Why Not Lower Score:**

This paper makes solid contribution to the multimodal pre-training community, and the general audience will be interested in this paper. It shows how to pre-train a large-scale multilingual multimodal model, and achieves strong performance. Though the technical novelty is rather limited, this paper still has high significance by showing the power of scaling up model training and how to scale it up.

**Metareview: Summary, Strengths And Weaknesses:**

This work introduces PaLI, a new large-scale vision-langauge model with an encoder-decoder architecture pre-trained over large-scale multilingual image-text pairs.

After author rebuttal, it received scores of 6888. All the reviewers are happy about the paper, agreeing that (1) results are strong and experiemnts are extensive, (2) sufficient in-depth analysis and ablation studies are performed for better model understanding, and (3) paper is well written. On the down side, the novelty of the model itself is kind of limited, and the reproducibility is low due to the nature of large-scale pre-training with in-house data.

Overall, this paper makes solid contributions and the general audience will be interested in this paper, therefore, the AC would like to recommend acceptance of the paper.

**Note From Pc:**

if the above contains the word "oral" or "spotlight" please see: "oral" presentation means -> notable-top-5% and "spotlight" means -> notable-top-25%. As stated in our emails, we are disassociating presentation type from AC recommendations